# Neurovascular sequestration in paediatric *P. falciparum* malaria is visible clinically in the retina

Valentina Barrera[1†]*, Ian James Callum MacCormick[1,2†], Gabriela Czanner[1,3], Paul Stephenson Hiscott[1], Valerie Ann White[4,5], Alister Gordon Craig[6], Nicholas Alexander Venton Beare[1,7], Lucy Hazel Culshaw[1], Yalin Zheng[1], Simon Charles Biddolph[8], Danny Arnold Milner[9], Steve Kamiza[10], Malcolm Edward Molyneux[2,6], Terrie Ellen Taylor[11,12], Simon Peter Harding[1,7]

[1]Department of Eye and Vision Science, Institute of Ageing and Chronic Disease, University of Liverpool, Liverpool, United Kingdom; [2]Malawi-Liverpool-Wellcome Trust Clinical Research Programme, College of Medicine, Blantyre, Malawi; [3]Department of Biostatistics, Institute of Translational Medicine, University of Liverpool, Liverpool, United Kingdom; [4]Department of Pathology and Laboratory Medicine, University of British Columbia and Vancouver General Hospital, Vancouver, Canada; [5]Department of Ophthalmology and Visual Science, University of British Columbia and Vancouver General Hospital, Vancouver, Canada; [6]Liverpool School of Tropical Medicine, Liverpool, United Kingdom; [7]St Paul's Eye Unit, Royal Liverpool University Hospital, Liverpool, United Kingdom; [8]National Specialist Ophthalmic Pathology Service, Royal Liverpool University Hospital, Liverpool, United Kingdom; [9]Center for Global Health, American Society for Clinical Pathology, Chicago, United States; [10]Department of Histopathology, College of Medicine, University of Malawi , Blantyre, Malawi; [11]Blantyre Malaria Project, College of Medicine, University of Malawi, Blantyre, Malawi; [12]Department of Osteopathic Medical Specialties, College of Osteopathic Medicine, Michigan State University, East Lansing, United States

*For correspondence:
v.barrera@liverpool.ac.uk

[†]These authors contributed equally to this work

Competing interests: The authors declare that no competing interests exist.

**Abstract** Retinal vessel changes and retinal whitening, distinctive features of malarial retinopathy, can be directly observed during routine eye examination in children with *P. falciparum* cerebral malaria. We investigated their clinical significance and underlying mechanisms through linked clinical, clinicopathological and image analysis studies. Orange vessels and severe foveal whitening (clinical examination, n = 817, OR, 95% CI: 2.90, 1.96–4.30; 3.4, 1.8–6.3, both p<0.001), and arteriolar involvement by intravascular filling defects (angiographic image analysis, n = 260, 2.81, 1.17–6.72, p<0.02) were strongly associated with death. Orange vessels had dense sequestration of late stage parasitised red cells (histopathology, n = 29; sensitivity 0.97, specificity 0.89) involving 360° of the lumen circumference, with altered protein expression in blood-retinal barrier cells and marked loss/disruption of pericytes. Retinal whitening was topographically associated with tissue response to hypoxia. Severe neurovascular sequestration is visible at the bedside, and is a marker of severe disease useful for diagnosis and management.
DOI: https://doi.org/10.7554/eLife.32208.001

## Introduction

Paediatric cerebral malaria (CM) is a frequently fatal complication of *Plasmodium falciparum* malaria that disproportionately afflicts children in sub-Saharan Africa; the WHO Malaria Report estimated that malaria killed 429,000 people worldwide in 2016, about 70% of whom were African children under 5 years of age (World Malaria Report 2016). CM is clinically defined as peripheral parasitaemia with coma not directly attributable to convulsions, hypoglycaemia, meningitis or any other identifiable cause (*Newton et al., 1998*). This definition is broad and is likely to over-diagnose a significant proportion of cases. The presence of a retinopathy known as malarial retinopathy (MR), and described by us with other colleagues (*Lewallen et al., 1993*; *Hero et al., 1997*; *Beare et al., 2004*; *Harding et al., 2006*), increases specificity when included in the diagnostic criteria (*Taylor et al., 2004*; *Beare et al., 2011*; *Barrera et al., 2015*).

Sequestration of parasitised red blood cells (pRBC) in the cerebral neurovasculature is the key underlying pathophysiological feature in *P. falciparum* CM (*Turner, 1997*). Unlike in the brain, the degree and location of neurovascular abnormalities can be observed clinically in the retina using routinely available ophthalmological techniques (*Harding et al., 2006*). Features comprise orange or white retinal vessels, patchy or confluent retinal whitening and white centred retinal haemorrhages (*Figure 1*). Severity of MR predicts the risk of death and duration of coma (*Beare et al., 2004*; *Lewallen et al., 2008*; *Beare et al., 2006*).

The management of *P. falciparum* malaria is changing. The incidence has fallen but is notoriously difficult to enumerate. Clearly, malaria is still causing significant numbers of deaths each year despite widespread use of artesunate-based combination therapies and moves to improve the early diagnosis of CM in district general hospitals (World Malaria Report 2016). New diagnostic and therapeutic interventions are being developed and tested. Our group has developed an automated algorithm platform for detection of MR from colour photographs (*Joshi et al., 2017*).

We with other colleagues have previously reported descriptive pathological investigations of the features of MR (*Lewallen et al., 2000*), including clinical associations (*White et al., 2009*) and suggesting mechanisms. We have previously hypothesised that the orange vessels in the retina (*Lewallen et al., 2000*) and the intravascular material seen on fluorescein angiography may indicate sequestration (*Beare et al., 2009*), but definitive evidence is required. We have previously identified that retinal whitening is caused by capillary nonperfusion, but the relationship of this nonperfusion to sequestration is unclear.

We studied orange and white vessels and retinal whitening to understand sequestration and its effects on the retina, and to inform clinical management of CM. We addressed these complex questions in a large series of children with CM recruited over 15 years all of whom had retinal examinations (clinical dataset), and in two subgroups, one comprising children who died and from whom eyes were available for histopathology (clinicopathology dataset) and a second of children who underwent retinal angiography (image analysis dataset). Findings from other cohorts and subcohorts from our programme have been reported previously by our group, addressing other research questions. The further analysis of our clinical dataset is an extension of our previous association study, while all other analyses presented in this manuscript are new.

## Results

### Correlation of vessel discolouration with disease outcome (clinical dataset)

We investigated the clinical significance of orange vessels seen in children admitted between 1999 and 2014 who had a retinal examination within 24 hr of admission and who were retinopathy-positive. Representative clinical photographs are given in *Figure 1*. *Figure 1—figure supplement 1*, shows the patient allocation of 1684 children admitted to the paediatric research ward.

The groups of subjects who did (n = 1160) and did not (n = 515) have an admission retinal exam were compared to assess possible selection bias (*Supplementary file 1*). Subjects who did not have an admission retinal exam were likely to have a higher serum lactate concentration (p<0.001) and were more likely to die (p<0.006). They were, on average, 5 months younger (p<0.001) and 0.2 kg lighter (p<0.01) than those who had retinal examinations.

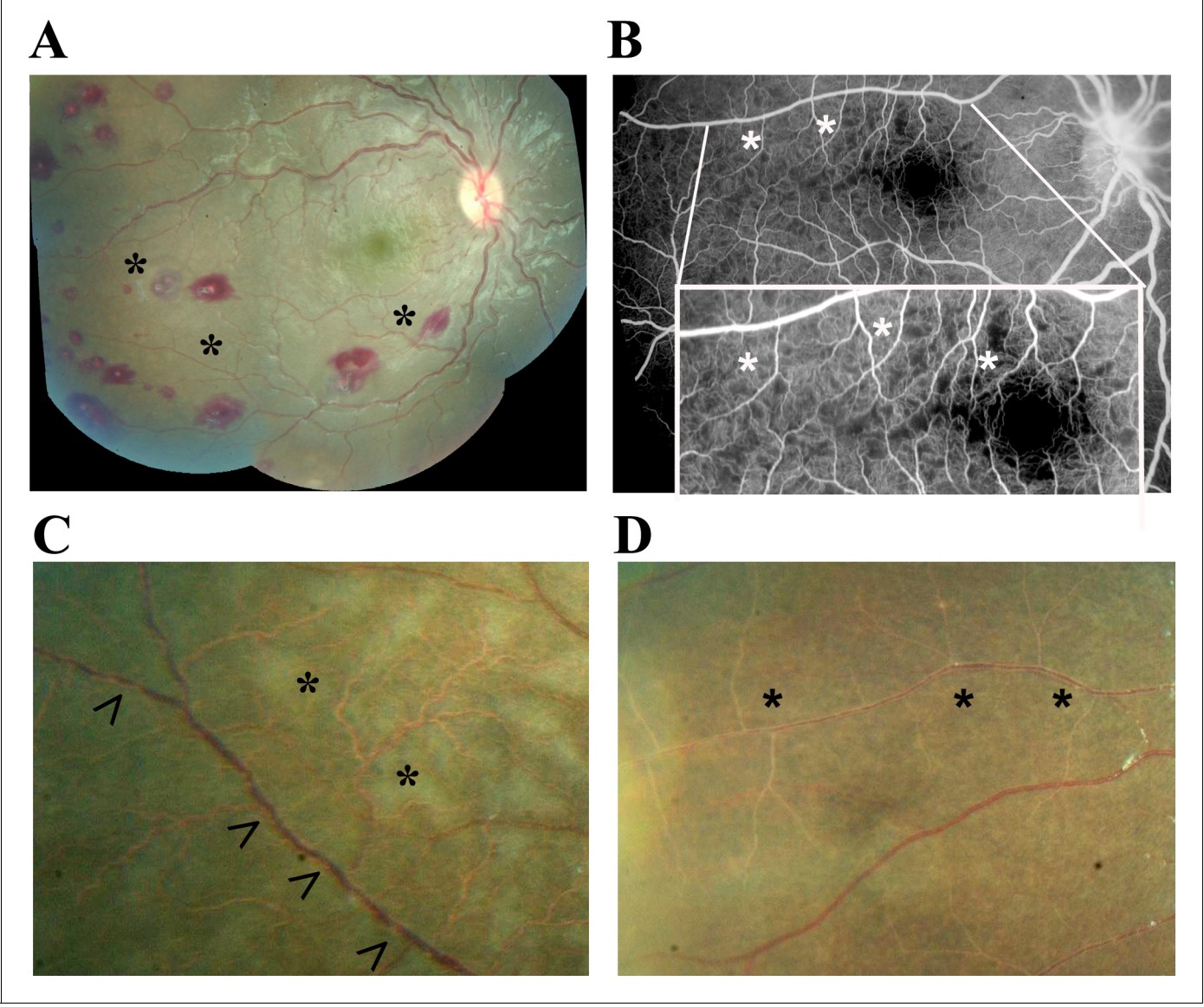

**Figure 1.** Principal features of malarial retinopathy (MR). (**A**) Montage image showing MR pathological features, including orange vessels (asterisks), white centred haemorrhages and whitening. (**B**) Corresponding fluorescein angiogram showing capillary nonperfusion (asterisks) mapping to retinal whitening. (**C–D**) Colour fundus image of retinopathy positive eyes (C, right; D, left eye; eyes were from different cases) showing orange intravascular material in large (arrowheads), small and postcapillary venules (asterisks), and capillaries; note retinal whitening also present.

DOI: https://doi.org/10.7554/eLife.32208.002

The following figure supplement is available for figure 1:

**Figure supplement 1.** Flow chart describing clinical dataset.

DOI: https://doi.org/10.7554/eLife.32208.003

On admission, 817 subjects had retinopathy-positive CM. Of these, 137 (16.8%) died with the time from admission to death less than 24 hr for the majority. In 663 subjects, data were available recording the time taken to recover consciousness, and of these 200 (30.2%) reached Blantyre Coma Score (BCS) $\geq$3/5 within 12 hr, 214 (32.3%) did so between 12 and 24 hr, and 249 (37.6%) took over 24 hr. Missing data were low at <10% for most variables apart from: blood lactate (~20%), HIV status (15%) and disc hyperaemia (12%).

Unadjusted associations between the presence and severity of clinical ophthalmoscopic features (*Figure 1*) and death in n = 817 with MR-positive CM, and admission eye examination, are shown in *Table 1*. Papilloedema (odds ratio (OR) 2.29, 95% confidence interval 1.55–3.38, p<0.001) and disc

**Table 1.** Associations with death in 817 subjects with admission retinal exam and retinopathy-positive paediatric cerebral malaria, 137 of whom died and 680 survived.

Retinal features are presented for the worse eye. Estimates are from unadjusted logistic regression. p≤0.01 is bold.

| Variable name | Units | Died Numerical characteristics | | | Survived Numerical characteristics | | | Association with death OR | 95% CI | p |
|---|---|---|---|---|---|---|---|---|---|---|
| Demographics | | | | | | | | | | |
| Age (median, IQR) | months | 35 | 23–59 | 136 | 39 | 27–58.75 | 680 | 0.99 | 0.00–1.00 | 0.43 |
| Weight (median, IQR) | kg | 11 | 9–15 | 137 | 12 | 10–15 | 680 | 0.97 | 0.93–1.02 | 0.22 |
| Height (median, IQR) | cm | 89 | 79–103 | 135 | 92 | 83–103 | 671 | 0.99 | 0.98–1.00 | 0.15 |
| Sex (%) | boy | 48.9 | | 66 | 50.29 | | 680 | 1.06 | 0.73–1.53 | 0.77 |
| | girl | 51.1 | | 69 | 49.71 | | | | | |
| Clinical | | | | | | | | | | |
| Coma score (%) | 0 | 23.3 | | 32 | 9.85 | | 67 | 3.57 | 2.13–5.88 | **<0.001** |
| | 1 | 41.6 | | 57 | 37.7 | | 256 | 2.13 | 1.28–3.57 | **0.003** |
| | 2 | 35.0 | | 48 | 52.5 | | 357 | reference | | |
| Respiratory distress (%) | Present | 48.9 | | 67 | 39.0 | | 265 | 1.5 | 1.04–2.17 | 0.03 |
| | Absent | 51.1 | | 70 | 61.0 | | 415 | | | |
| Convulsions at admission (%) | Present | 12.4 | | 17 | 14.9 | | 98 | 0.83 | 0.45–1.44 | 0.51 |
| | Absent | 87.6 | | 120 | 85.4 | | 574 | | | |
| Temperature (median, IQR) | degrees C | 38.7 | 37.8–39.5 | 137 | 38.9 | 38–39.7 | 680 | 0.89 | 0.77–1.03 | 0.12 |
| Systolic BP (median, IQR) | mmHg | 100 | 90–110 | 127 | 100 | 90–110 | 652 | 0.99 | 0.99–1.01 | 0.63 |
| Pulse (median, IQR) | beats/min | 156 | 136.5–170.5 | 137 | 152 | 136.75–169 | 678 | 1.0 | 0.99–1.01 | 0.98 |
| Duration of coma (median, IQR) | Hours | 7 | 4–18 | 110 | 7 | 4–17 | 558 | 0.99 | 0.98–1.01 | 0.29 |
| Duration of fever (median, IQR) | Hours | 48 | 33.25–72 | 130 | 60 | 43.25–72 | 652 | 0.99 | 0.99–1.00 | 0.09 |
| Hypoglycaemia on ward (%) | Present | 14.6 | | 20 | 7.81 | | 53 | 2.02 | 1.16–3.5 | 0.012 |
| | Absent | 85.4 | | 117 | 92.1 | | 626 | | | |
| Laboratory | | | | | | | | | | |
| Parasitaemia (median, IQR) | #cells | 79052 | 16695–357000 | 134 | 68076 | 11700–298000 | 649 | 1.0 | 0.99–1.00 | 0.27 |
| White cell count (median, IQR) | #cells | 11300 | 6925–18225 | 120 | 9200 | 6600–13725 | 630 | 1.0 | 1.00–1.00 | **0.004** |
| Haematocrit (median, IQR) | % | 19.5 | 15–24.75 | 136 | 20 | 15.8–25 | 673 | 0.99 | 0.97–1.02 | 0.69 |
| Lactate (median, IQR) | mmol/L | 8.75 | 5.38–12.78 | 92 | 5.3 | 3.2–9.9 | 519 | 1.11 | 1.06–1.16 | **<0.001** |
| HRP2 (median, IQR) | ng/ml | 8838.5 | 4435.5–15102.3 | 120 | 5765 | 2471.5–10031 | 609 | 1.0 | 1.00–1.00 | **0.004** |
| HIV (%) | Positive | 22.5 | | 29 | 14.9 | | 88 | 1.66 | 1.03–2.66 | 0.036 |
| | Negative | 77.5 | | 100 | 85.1 | | 503 | | | |
| Ophthalmoscopy | | | | | | | | | | |

*Table 1 continued on next page*

*Table 1 continued*

| Variable name | Units | Died Numerical characteristics | | Survived Numerical characteristics | | Association with death OR | 95% CI | p |
|---|---|---|---|---|---|---|---|---|
| Retinal haemorrhage (%) | >50 | 16.0 | 22 | 4.7 | 32 | 3.4 | 1.78–6.5 | **<0.001** |
| | 21 to 50 | 11.0 | 15 | 6.50 | 44 | 1.69 | 0.85–3.34 | 0.14 |
| | 6 to 20 | 13.1 | 18 | 19.0 | 129 | 0.69 | 0.38–1.27 | 0.23 |
| | 1 to 5 | 32.9 | 45 | 42.9 | 291 | 0.76 | 0.48–1.23 | 0.27 |
| | None | 27.0 | 37 | 27.0 | 183 | reference | | |
| Macular whitening (%) | >1 | 23.9 | 32 | 14.8 | 100 | 2.31 | 1.16–4.59 | 0.017 |
| | 1/3 to 1 | 28.4 | 38 | 25.1 | 170 | 1.61 | 0.83–3.12 | 0.16 |
| | <1/3 | 37.3 | 50 | 45.2 | 306 | 1.18 | 0.63–2.22 | 0.61 |
| | None | 10.5 | 14 | 14.9 | 101 | reference | | |
| Foveal whitening (% of foveal zone) | >2/3 | 23.3 | 31 | 11.5 | 78 | 3.39 | 1.83–6.26 | **<0.001** |
| | 1/3 to 2/3 | 18.1 | 24 | 15.2 | 103 | 1.99 | 1.05–3.74 | 0.03 |
| | <1/3 | 42.8 | 57 | 46.8 | 316 | 1.54 | 0.90–2.62 | 0.11 |
| | none | 15.8 | 21 | 26.5 | 179 | reference | | |
| Temporal whitening (%) | 3 | 10.0 | 13 | 12.9 | 87 | 0.83 | 0.41–1.66 | 0.60 |
| | 2 | 24.6 | 32 | 18.4 | 124 | 1.43 | 0.83–2.47 | 0.20 |
| | 1 | 41.5 | 54 | 43.1 | 290 | 1.03 | 0.64–1.67 | 0.89 |
| | none | 23.9 | 31 | 25.6 | 172 | reference | | |
| Orange vessels, temporal quadrant (%) | present | 44.6 | 58 | 21.7 | 145 | 2.9 | 1.96–4.3 | **<0.001** |
| | absent | 55.4 | 72 | 78.3 | 523 | | | |
| White vessels, temporal quadrant (%) | present | 25.4 | 33 | 24.3 | 162 | 1.06 | 0.69–1.64 | 0.78 |
| | absent | 74.6 | 97 | 75.8 | 506 | | | |
| White capillaries (%) | present | 26.9 | 35 | 33.1 | 221 | 0.75 | 0.49–1.13 | 0.17 |
| | absent | 73.1 | 95 | 66.9 | 447 | | | |
| Papilloedema (%) | present | 39.0 | 53 | 21.8 | 148 | 2.29 | 1.55–3.38 | **<0.001** |
| | absent | 61.0 | 83 | 78.2 | 530 | | | |
| Disc hyperaemia (%) | present | 48.7 | 54 | 35.3 | 212 | 1.73 | 1.15–2.61 | **0.008** |
| | absent | 51.4 | 57 | 64.7 | 388 | | | |

DOI: https://doi.org/10.7554/eLife.32208.004

hyperaemia (OR 1.73, 1.15–2.62, p<0.01), both indicators of brain swelling, were more likely in those who died. White cell count and blood HRP2 concentration had statistically significant associations with death, but with very small effect sizes (OR very close to 1).

The presence of visible orange vessels on ophthalmoscopy (*Figure 1C–D*) was significantly associated with death (OR 2.90, 1.96–4.30, p<0.001), as was severe foveal whitening (>2/3 foveal area; OR 3.40, 1.80–6.30, p<0.001; simple logistic regression; *Table 1*). When including potential confounders (age, WCC, HRP2, lactate, papilloedema - see Materials and methods) in a multivariable regression model for the presence of the two retinal features with death, we found similar ORs and significance (orange vessels: OR 2.85, 1.72–4.74, p<0.001, n = 549; foveal whitening: OR 3.57, 1.57–8.13, p=0.002, n = 615).

## Clinicopathological characterisation of retinal intravascular material (clinicopathology dataset)

Twenty-nine cases from the autopsy archive met the inclusion criteria for our clinicopathological study of the nature and effects of retinal intravascular material; details of the dataset are given in *Table 2*, and records of pre-mortem retinal clinical examination in *Table 3*. Of these cases, 21 had

**Table 2.** Summary of clinicopathology dataset.

| Clinicopathological investigation (per MR feature) | Number of cases analysed | Number of retinal layers analysed | Number of vessels counted |
| --- | --- | --- | --- |
| *Vessel changes (H and E; GFAP; FGN; ICAM-1)* | | | |
| PO block analysis | 27 | – | 100 |
| Calotte analysis | 6 | – | 100 |
| Punch biopsies | 4 | – | 50 |
| *Retinal whitening (VEGFR1; AQP4)* | | | |
| Macular analysis | 20 | 4 | – |
| Peripheral retinal analysis | 21 | 4 | – |

DOI: https://doi.org/10.7554/eLife.32208.005

MR (Grade 1 n = 5, Grade 2 n = 16). In all MR-positive cases, intracerebral and intraretinal sequestration of parasitised red blood cells (pRBC) post mortem exceeded 23% of capillaries and venules, consistent with a histological diagnosis of CM (*Taylor et al., 2004*; *Barrera et al., 2015*). Autopsy confirmed a cause of death different from CM in the eight MR-negative control patients (Grade 0). Twelve out of the 29 autopsy cases were HIV-positive.

We investigated the nature of the intravascular material identifiable clinically and pathologically, primarily by colour changes in venules and capillaries, in 12 out of 21 MR-positive patients (*Figure 2A–B*). Intravascular filling defects (IVFD) within the blood column were identified in retinal venules on all the five cases with fluorescein angiography (FA) available. Orange and white microvessels (cases = 12, vessels = 212) were sampled using manual microdissection techniques (marked quadrant, *Figure 2A*), and compared microscopically with clinically normal vessels (cases = 8, vessels = 200) in different retinal segments of the same case or from different specimens across Grade 1 and Grade 2 MR groups. All orange vessels exhibited pigmented pRBCs sequestered in layers on the endothelium at the margin of the vessel lumen, with a blood column consisting of uninfected RBCs in the centre of affected vessels (*Figure 2C*, *Figure 3*). These vessels were occasionally surrounded by extravasated RBCs in the absence of clinically visible haemorrhage. White vessels (usually distended capillaries) contained primarily extraerythrocytic haemozoin (HZ) and some remnants of pRBC; non-parasitised RBCs were absent. Fibrin polymers were detected in retinal capillaries and venules (*Figure 3—figure supplement 1*). All vessels that appeared normal, during clinical and gross examination, lacked these features (*Figure 2D*).

H&E analysis of orange intravascular material (n = 3 cases) showed aggregates containing both abundant sequestered pigmented (late stage) pRBCs and non-parasitised RBCs in venules (*Figure 2C*, *Figure 3A–B*). These clusters of pRBC were not observed in vessels without FA filling defects from the remaining two MR-positive cases for which FA was available.

We investigated the relationship between severity/extent of late stage pRBCs and presence of visible orange discolouration in nine MR-positive cases (n = 412 vessels studied; *Table 4*). Vessels with sequestered late stage pRBCs involving 360° of the circumference of the vessel lumen were strongly associated with the presence of orange discoloration (*Table 4*). Sensitivity and specificity for orange discolouration as an indicator of this extent of sequestration were 0.97 (95% confidence interval: 0.94 to 0.99) and 0.89 (0.84 to 0.93), respectively, with positive and negative predictive values of 0.88 (0.83 to 0.92) and 0.98 (0.94 to 0.99).

## Tissue effects of retinal neurovascular sequestration

We studied the effects of pRBC sequestration on cellular vessel wall components in MR-positive and negative cases, in vessels with and without sequestered pRBCs in matched tissue sections assessing presence/absence of continuous (annular) immunostaining. In both Grades 1 and 2 MR-positive cases, intraretinal sequestration was significantly associated with reduced expression in retinal microvessels of the endothelial cell membrane glycoprotein CD34, the pericyte structural protein smooth muscle actin (SMA) and the signalling molecule platelet derived growth factor receptor β (PDGFRβ) (*Figure 4*, all p<0.005); SMA was only reported for venules as it does not produce an annular staining pattern in normal capillaries. The proportions of continuous immunostaining in capillaries and venules, with and without pRBC sequestration, respectively (means ±SD), were: CD34, 14 ± 9% and

**Table 3.** Retinal pathological features and scores for 29 study subjects in the clinicopathology dataset

| Case n. | MR* Grade | Eye[†] | Vessel changes (Q)[‡] | Vessels[§] | Localization[#] | Haem[¶] | Macular whitening[¥] | Central retinal whitening (overall score)** | Peripheral whitening (score) | Whitening: retinal quadrants | Papill-oedema[h††] (score) |
|---|---|---|---|---|---|---|---|---|---|---|---|
| 1 | 2 | RE | 4 Q | Ven + Cap | All quadrants | >50 | 1/3–1 DA | 4 | 3 | 4 Q | 2 |
| 2 | 2 | RE | 4 Q | Ven + Cap | All quadrants | 1–5 | ≥1 DA | 6 | 3 | 4 Q | 2 |
| 3 | 2 | LE | 4 Q | Ven | All quadrants | 1–5 | ≥1 DA | 6 | 1.75 | 4 Q | 2 |
| 4 | 2 | RE | 4 Q | Ven | All quadrants | >50 | 1/3–1 DA | 5 | 1.5 | T + N | 0 |
| 5 | 2 | RE | 3 Q | Ven + Cap | T + N + S | 1–5 | ≥1 DA | 6 | 2.7 | T + N + S | 0 |
| 6 | 2 | RE | 2 Q | Ven | T + S | >50 | <1/3 DA | 2 | 0.75 | T + S | 2 |
| 7 | 2 | LE | None | None | 0 | 6–20 | ≥1 DA | 6 | 0.25 | T | 2 |
| 8 | 2 | RE | None | None | 0 | 0 | ≥1 DA | 6 | 2 | 4 Q | 2 |
| 9 | 2 | LE | 4 Q | Ven + Cap | All quadrants | 0 | ≥1 DA | 6 | 1 | 4 Q | 0 |
| 10 | 2 | LE | None | None | 0 | 21–50 | 1/3–1 DA | 4 | 1.5 | 4 Q | 0 |
| 11 | 2 | LE | 3 Q | NA | NA | 0 | ≥1 DA | 4 | 2 | 4 Q | 0 |
| 12 | 2 | LE | None | None | 0 | 6–20 | 1/3–1 DA | 4 | 0 | 0 | 2 |
| 13 | 2 | LE | None | None | 0 | 1–5 | ≥1 DA | 6 | 1 | 4 Q | 0 |
| 14 | 2 | RE | NA | NA | NA | 1–5 | 1/3–1 DA | 4 | NA | NA | 2 |
| 15 | 2 | LE | 3 Q | Ven + Cap | T + N + S | 1–5 | <1/3 DA | 2 | 0.7 | T + N + S | 0 |
| 16 | 2 | LE | 3 Q | Ven | T + N + S | 1–5 | <1/3 DA | 2 | 0.5 | I + N | 0 |
| 17 | 1 | RE | 1 Q | None | 0 | 0 | <1/3 DA | 2 | 1 | T + S | 2 |
| 18 | 1 | RE | 1 Q | Cap | T | 0 | <1/3 DA | 2 | 1 | 4Q | 0 |
| 19 | 1 | RE | None | None | 0 | 1–5 | <1/3 DA | 2 | 1 | T + N | 0 |
| 20 | 1 | LE | None | None | 0 | 1–5 | <1/3 DA | 2 | 0 | NA | 0 |
| 21 | 1 | LE | None | None | 0 | None | None | 0 | 0.25 | 0 | 0 |
| 22 | 0 | RE | None | None | 0 | None | None | 0 | 0 | 0 | 0 |
| 23 | 0 | LE | None | None | 0 | None | None | 0 | 0 | 0 | 0 |
| 24 | 0 | RE | None | None | 0 | None | None | 0 | 0 | 0 | 0 |
| 25 | 0 | LE | None | None | 0 | None | None | 0 | 0 | 0 | 0 |
| 26 | 0 | LE | None | None | 0 | None | None | 0 | 0 | 0 | 0 |
| 27 | 0 | RE | None | None | 0 | None | None | 0 | 0 | 0 | 0 |
| 28 | 0 | LE | None | None | 0 | None | None | 0 | 0 | 0 | 0 |
| 29 | 0 | RE | None | None | 0 | >50 | None | 0 | 0 | 0 | 0 |

*MR = malarial retinopathy. Grade was defined based on percentage of retinal vessels with sequestration (**Beare et al., 2004**) as explained in Methods. Last peripheral parasitaemia (expressed as asexual pRBCs/µl blood, geometric means reported) was: 42,200 (Grade 0), 43,212 (Grade 1) and 9357 (Grade 2).

[†]Eye: RE = right eye; LE = left eye vessel changes:

[‡](Q)=number of retinal quadrants affected.

[§]Vessels: Ven = venules; Cap = capillaries.

[#]Localisation of vessel changes: I = inferior; N = Nasal; S = superior; T = temporal.

[¶]Haem = no. of retinal haemorrhages.

[¥]Extent of whitening is shown for macula in disc areas (DA).

**Central whitening (overall score)=sum of macular and foveal whitening scores assigned as: 1 =<1/3 DA or FA, 2 = 1/3–1 DA or 1/3-2/3FA, 3 =>1 DA or >2/3FA.

[††]Papilloedema is the swelling of optic disc caused by increased intracranial pressure. The significance of papilloedema in cerebral malaria is not clear; however, it is the strongest risk factor for poor outcome among comatose children with clinical cerebral malaria.

DOI: https://doi.org/10.7554/eLife.32208.006

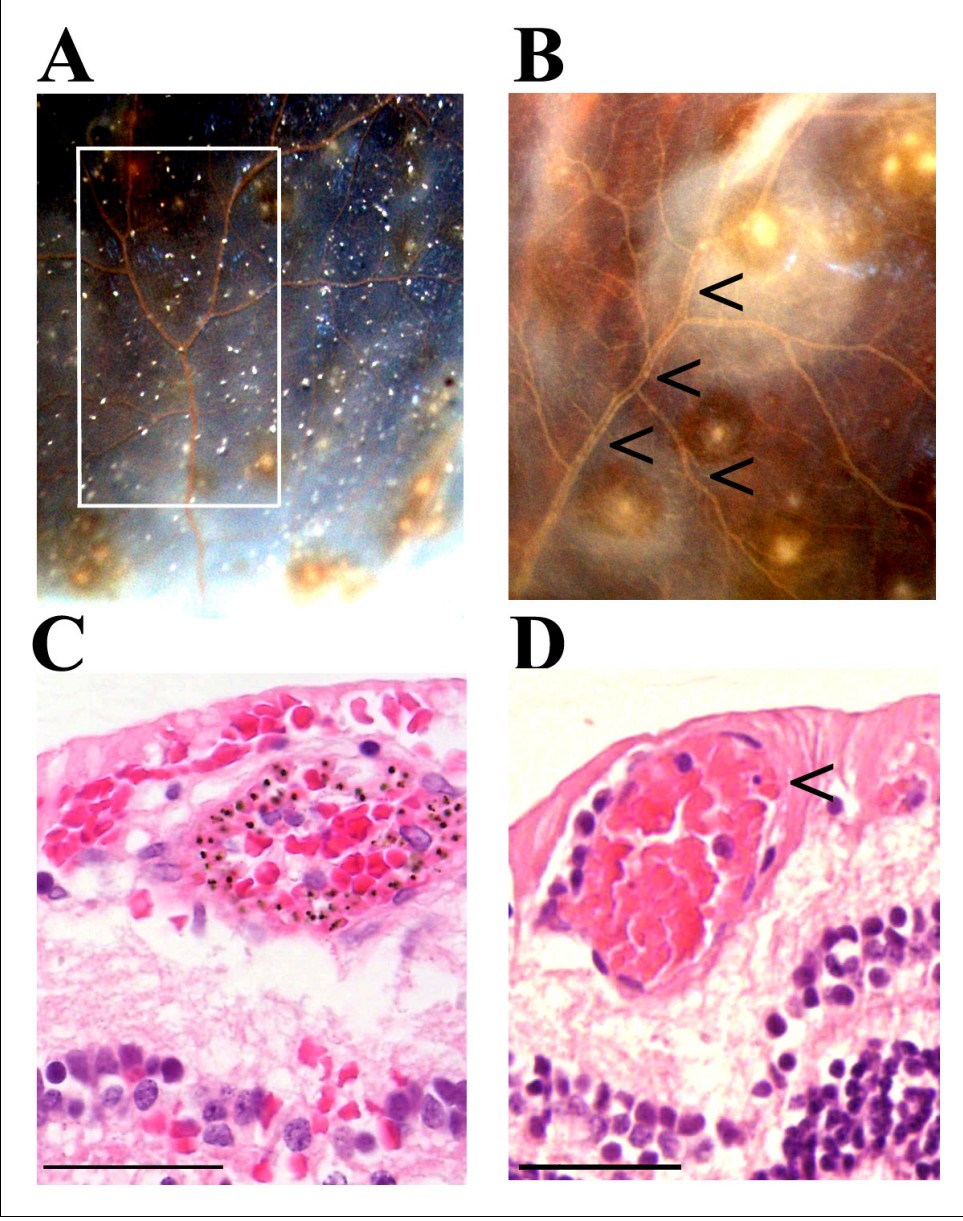

**Figure 2.** Vessel changes in malarial retinopathy. (**A–B**) Vessel colour changes (panels A-B) and intravascular filling defects (panel B, arrowheads) were identified during gross pathology examination (representative images of superior calotte and PO block from histology cases n. 5 and 7, respectively) N = 12. Abnormal vessels were sampled during gross pathology examination and analysed separately (see marked quadrant in panel A). (**C–D**) H and E staining of parasitised venules from MR cases sampled by punch biopsies from a retinal quadrant with (panel **C** shows the same orange vessel as in panel A) and without (panel D, case n. 15) vessel discolouration. (**C**) The margin of the vessel lumen has a near-complete layer of pigment-containing pRBCs (that stain less intensely pink than the adjacent non-parasitised RBC) on the endothelium. (**D**) Mild sequestration of pRBCs which is marked by an arrowhead. Scale bars (50 μm, (**C–D**).
DOI: https://doi.org/10.7554/eLife.32208.007

90 ± 10%; SMA 11 ± 10% and 65 ± 25%; PDGFRβ 19 ± 15% and 77 ± 18% (all p<0.005). These findings are consistent with marked altered cell function or loss of pericytes and endothelial cells of vessels with pRBC sequestration. To explore the impact of pericyte dysfunction on vessel stability, we tested for an association between reduced immunostaining and presence of retinal haemorrhages.

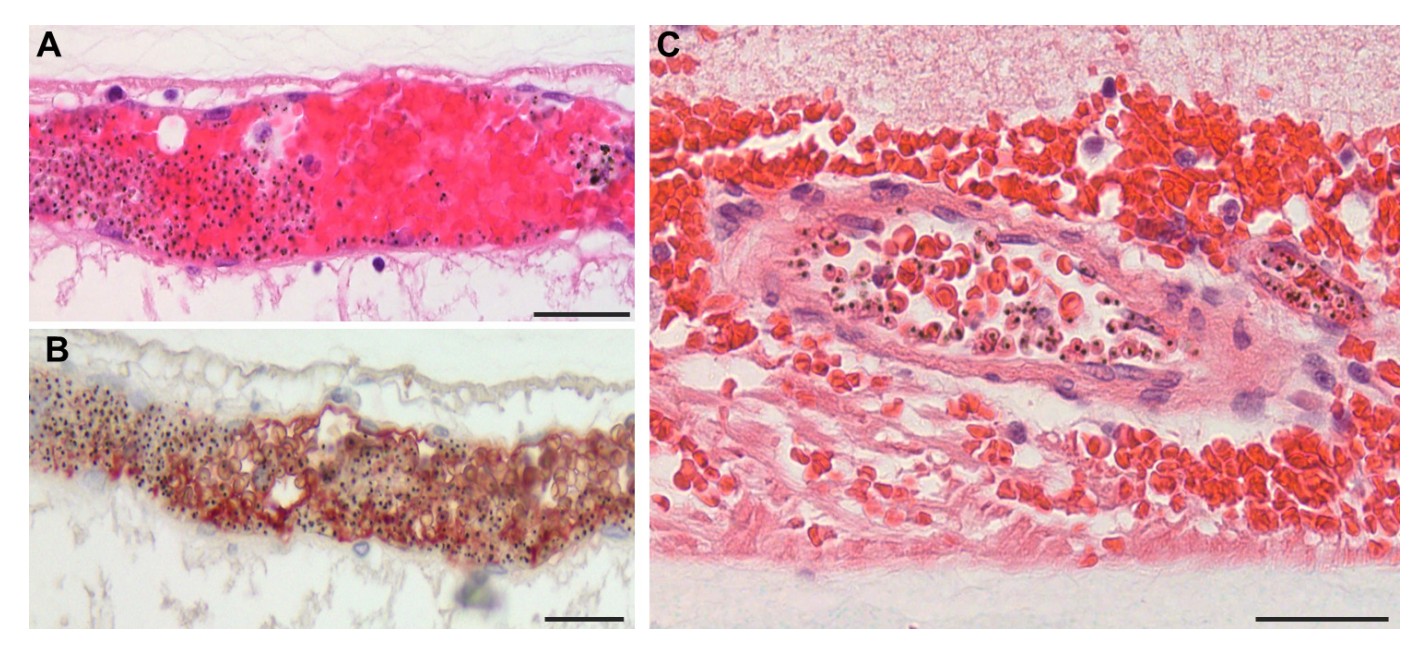

**Figure 3.** Severe pRBC sequestration in large venules and arterioles in MR with visible vessel discolouration. (**A–B**) Longitudinal section of large retinal venule from retinal area affected by intravascular filling defects on fluorescein angiography (histopathology case no. 9) analysed by H&E staining (**A**) and anti-fibrinogen IHC (**B**). Clusters of pRBC are seen within the vessel lumen and attached to the wall. (**C**) Cross section of a large retinal arteriole with moderate pRBC sequestration (case n. 5). Arteriole is surrounded by haemorrhage, probably of a venular origin as arteriolar vessel wall appeared intact (in multiple sections). Scale bars: 50 µm (all panels).

DOI: https://doi.org/10.7554/eLife.32208.008

The following figure supplement is available for figure 3:

**Figure supplement 1.** Detection of thrombi in post-mortem retinal periphery using a combination of MSB staining (panels **A,B**; arrows: intravascular thrombi are stained bright pink), and anti-CD61 platelet marker immunostaining (panel **C**, red stained).

DOI: https://doi.org/10.7554/eLife.32208.009

Percentages of vessels with normal PDGFRβ staining were significantly less in MR-positive cases with haemorrhages (18%) than those without (39%; n = 21, p<0.05).

Glial cells (principally astrocytes and Müller cells) surrounding venules and capillaries affected by severe pRBC sequestration were studied in 10 of 21 (48%) MR-positive cases (*Figure 5A–D*). There were statistically significant increases in perivascular astrocyte intercellular adhesion molecule 1 (ICAM-1) (p=0.003) and Müller cell cytoskeletal component glial fibrillary acidic protein (GFAP) (p=0.034), markers for early (4–12 hr) and late (after 24 hr) glial activation, respectively (*Lee et al., 2000*; *Hiscott et al., 1984*). No MR-negative cases showed perivascular ICAM-1 or GFAP immunore-activity. ICAM-1 tissue staining was also associated with the presence of discoloured vessels (*Figure 5A*, all Fisher exact tests p<0.05), compared with normal vessels where ICAM-1 was restricted to the endothelium (*Figure 5B*). Müller cell GFAP immunoreactivity was observed in 8 of

**Table 4.** Relationship between severe sequestration (pigmented/late parasitised RBCs sequestered around 360° of the lumen circumference) and orange discolouration visible clinically and on gross pathology in 412 venules (diameter 10–50 µm) from nine cases

|  |  | Orange discolouration | |
| --- | --- | --- | --- |
|  |  | + | - |
| Severe sequestration | + | 188 | 5 |
|  | - | 24 | 195 |

DOI: https://doi.org/10.7554/eLife.32208.010

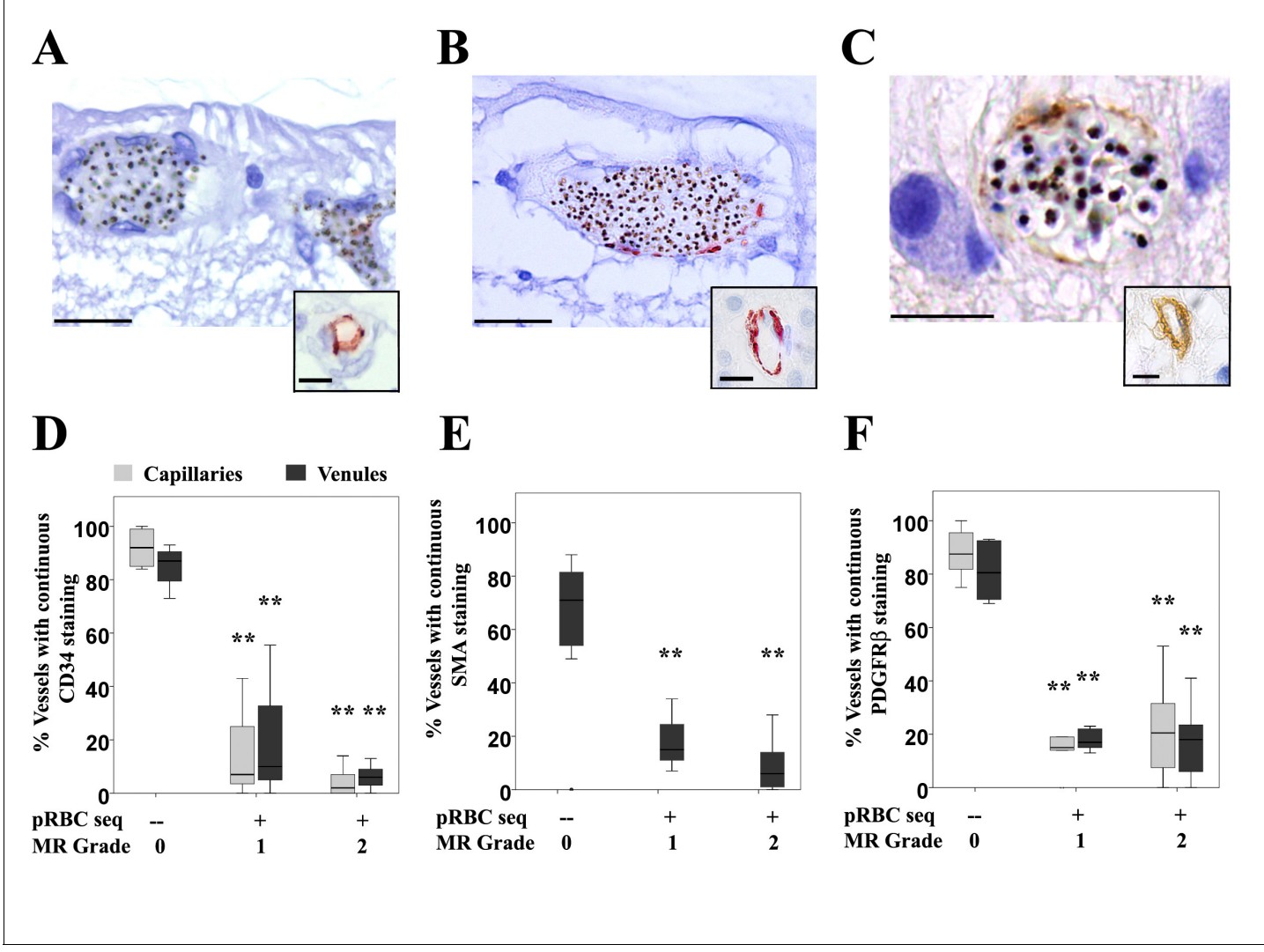

**Figure 4.** Vascular changes in retinal vessels in malarial retinopathy. (A–I) Expression of endothelial CD34 (panel A: case n. 3, inset: case n. 25 and D: box plot), pericytic SMA (panel B: case n. 12, inset: case n. 27 and E: box plot) and pericytic PDGFRβ (panel C: case n. 13, inset: case n. 26 and F: box plot) markers. Insets show normal annular staining in absence of pRBC sequestration, whereas this annular pattern is lost in the sequestrated vessels seen in A-C. SMA was only reported for venules as it does not produce an annular staining pattern in normal capillaries: panel E. N = 17 for CD34; N = 29 for SMA and PDGFRβ immunostaining. ANOVA was used to compare means. **$p<0.005$. Scale bars: 20 µm (A–C), 5 µm (insets).

DOI: https://doi.org/10.7554/eLife.32208.011

The following source data is available for figure 4:

**Source data 1.** Vascular changes in retinal vessels inmalarial retinopathy.

DOI: https://doi.org/10.7554/eLife.32208.012

the 21 (38%) cases with MR (*Figure 5C*) versus MR-negative cases (*Figure 5D*) where staining was restricted to the first retinal layer.

## Pathogenesis of retinal whitening

To test the hypothesis that retinal whitening is caused by hypoxia-induced cellular oedema (*Kaur et al., 2008a*), we compared the proportions and distribution of the tissue hypoxia and intra-cellular oedema markers VEGFR1 and AQP4, respectively (*Marti et al., 2000*; *Medana et al., 2011*), in MR-positive and negative cases.

MR-positive cases showed increased expression of VEGFR1 immunoreactivity in both central and peripheral retina. VEGFR1 immunostaining was primarily localised in the inner retina (*Figure 6A,B*)

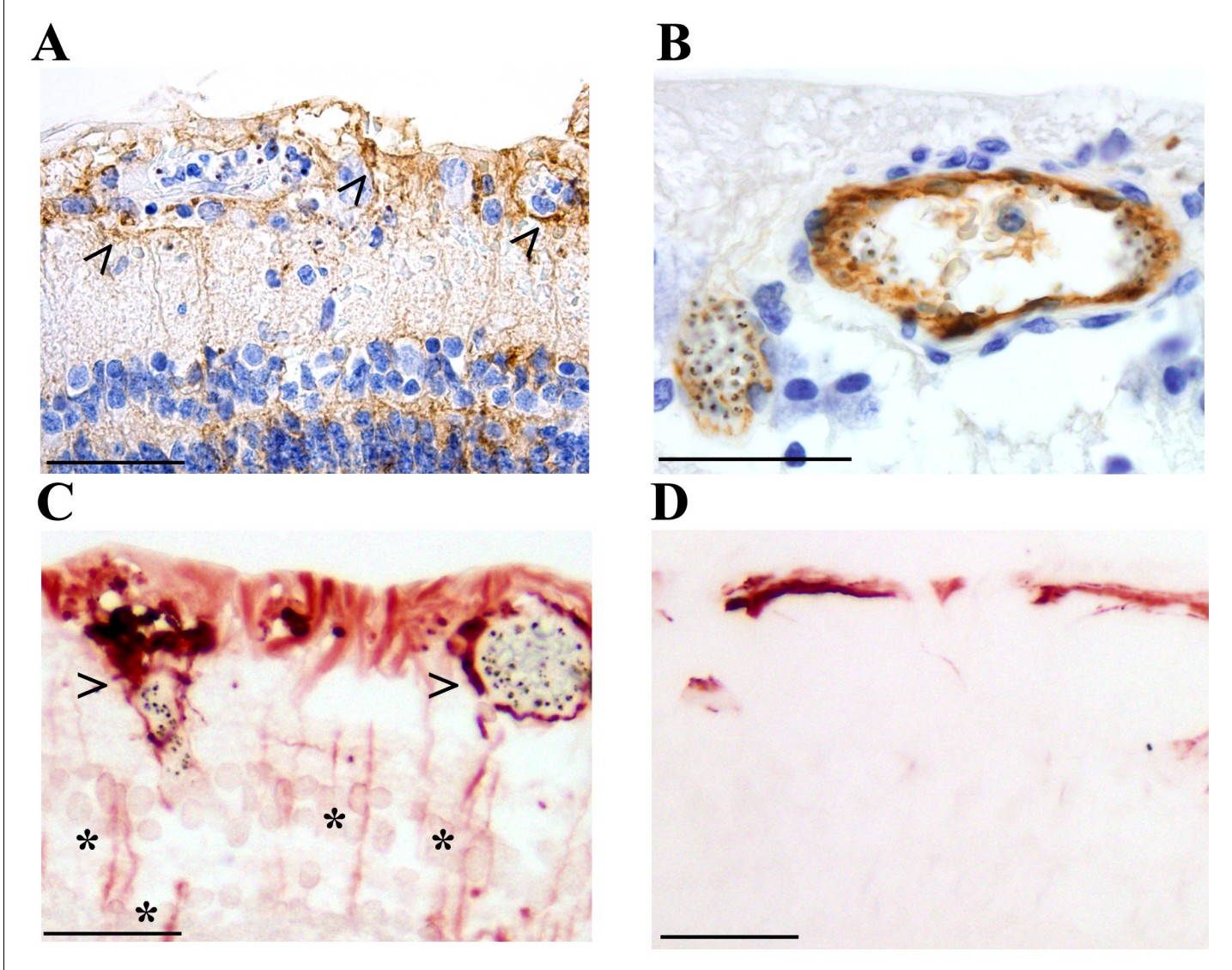

**Figure 5.** Activation of retinal glial cells in malarial retinopathy (MR). (**A–B**) Anti-ICAM-1 staining of MR-positive cases with (case n. 16, panel A) and without (case n. 13, panel B) vessel discolouration. Haematoxylin (blue) counterstain was used. (**C–D**) Anti-GFAP staining of orange-discoloured vessels in punch biopsy from MR-positive case n. 5, and in MR-negative case n. 25. Haematoxylin counterstaining was omitted here. In A and C, peri-vascular activated astrocytes and Műller cells are marked with arrowheads, and asterisks label Műller cell bodies. Scale bars: 50 μm (all panels).
DOI: https://doi.org/10.7554/eLife.32208.013

(ganglion cell (primarily in the macula) and inner nuclear cell bodies and synapses) and values were positively correlated with increasing severity of whitening for all these cells (*Figure 6C* $p < 0.05$, $p < 0.001$) and for macular ganglion cell layers with worse MR (*Figure 6D*, $p < 0.001$).

AQP4 expression levels were generally more intense in MR-positive cases with whitening (*Figure 1B*) than those without (*Figure 7*, *Figure 7—figure supplement 1*). High AQP4 staining levels were found in glial cells, including Műller cells, in the nerve fibre layer (NFL) and outer plexiform layer (*Chen et al., 2012*) in the macula (*Figure 7A,B*) and temporal periphery (*Figure 7—figure supplement 1A,B*). Densitometry analysis showed significantly higher AQP4 levels for macula and temporal periphery (*Figure 7C* and *Figure 7—figure supplement 1*, ANOVA test, $p < 0.05$ except moderate whitening). There were statistically significant associations also between AQP4 staining pattern and MR grade (*Figure 7D* and *Figure 7—figure supplement 1*). In addition to the association found between tissue whitening, VEGFR1 and AQP4 expression levels, in 44% and 68% of MR-

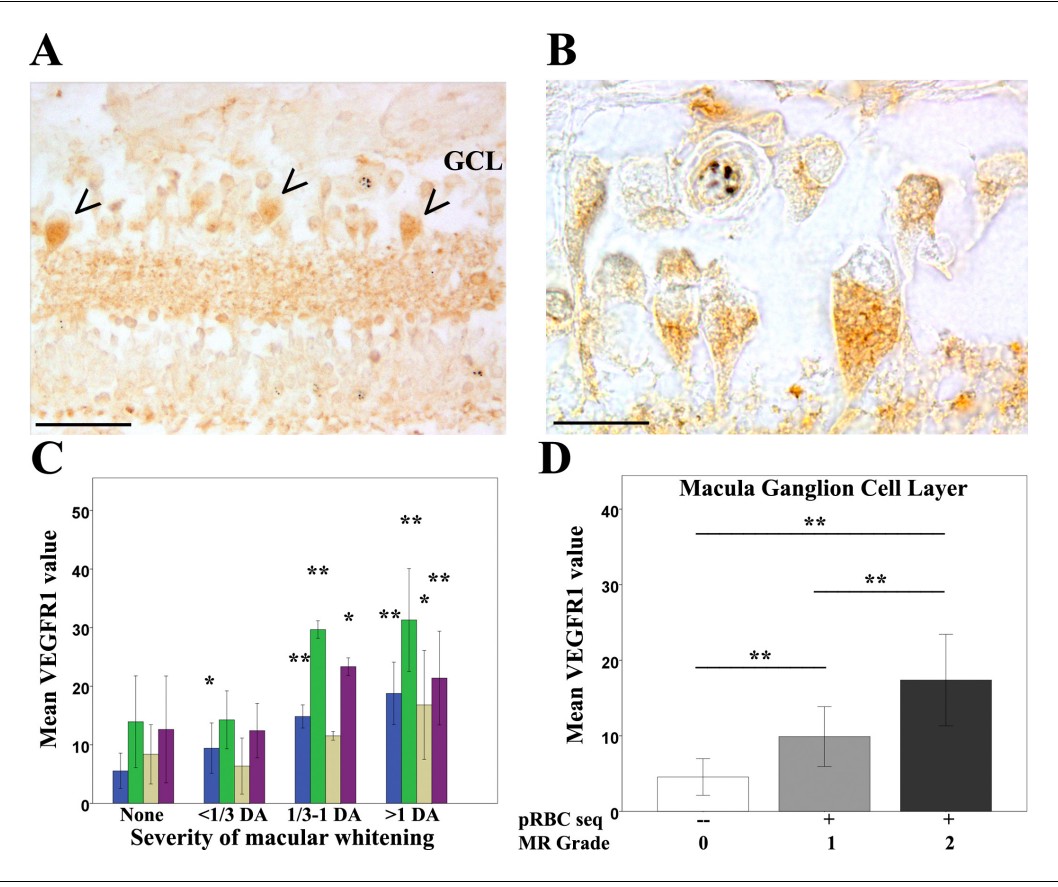

**Figure 6.** Clinicopathological association between retinal whitening in the macula and increased VEGFR1 expression in malarial retinopathy. (A–B) Immunostaining pattern in macula affected by whitening (case no 9) (low (A) and high (B) magnification; VEGFR1 +ve ganglion cell bodies indicated by arrowheads). (C) Cluster column chart showing densitometrically assessed intensity of immunoreactivity ('value') of VEGFR1 expression plotted by retinal layer against whitening severity, compared with MR –ve cases. Ganglion cell layer = GCL (blue); inner plexiform layer = IPL (green); inner nuclear layer = INL (light brown); outer plexiform layer = OPL (purple). (D) VEGFR1 levels in the GCL plotted against MR severity classification groups (grade 0 = none, 1 = mild, two moderate/severe). Means ± SD are reported in both charts; ANOVA was used to compare means (N = 26). *p≤0.05 and **p≤0.001. Scale bars: 50 μm (panel A); 20 μm (panel B).
DOI: https://doi.org/10.7554/eLife.32208.014

The following source data is available for figure 6:

**Source data 1.** Clinicopathological association between retinal whitening in the macula and increased VEGFR1 expression inmalarial retinopathy.
DOI: https://doi.org/10.7554/eLife.32208.015

positive cases (macula and periphery respectively) intravascular thrombi were co-localised with retinal whitening (*Figure 3—figure supplement 1A–C*; p<0.05 for periphery only).

## Fluorescein angiography and image analysis study of retinal sequestration (image analysis dataset)

Between 2009 and 2014, 260 subjects with MR-positive CM underwent retinal FA on the day or day after admission. A representative FA of IVFD is shown in *Figure 1B*, with the dataset in *Figure 8* and the rates and location of IVFD in *Table 5*. The topographical correlation between ophthalmoscopic and angiographic features of IVFD is illustrated in *Figure 9*. IVFD occurred frequently in the retinal venules (large 80.2%, small 98.0%, post capillary 98.3%). There was no association between sequestration in post-capillary venules and survival (OR 0.23, 0.054–1.02, p=0.053). Conversely, sequestration was infrequent in the arterioles but with significant associations with death for large arteriole

sequestration (OR 2.81, 1.17–6.72, p<0.02), and non-significant association for precapillary arterioles (OR 2.47, 0.94–6.45, p=0.065) (see *Table 5* and *Figure 9*). Similar findings were found for time to recovery of consciousness (binomial regression coefficient, 95% CI): precapillary arterioles (0.32, 0.094–0.55, p<0.01), small arterioles (0.30, 0.093–0.51, p<0.01), large arterioles (0.38, 0.076–0.68, p<0.02). Sequestration in the capillaries was frequently seen but was ungradeable in 62% of cases.

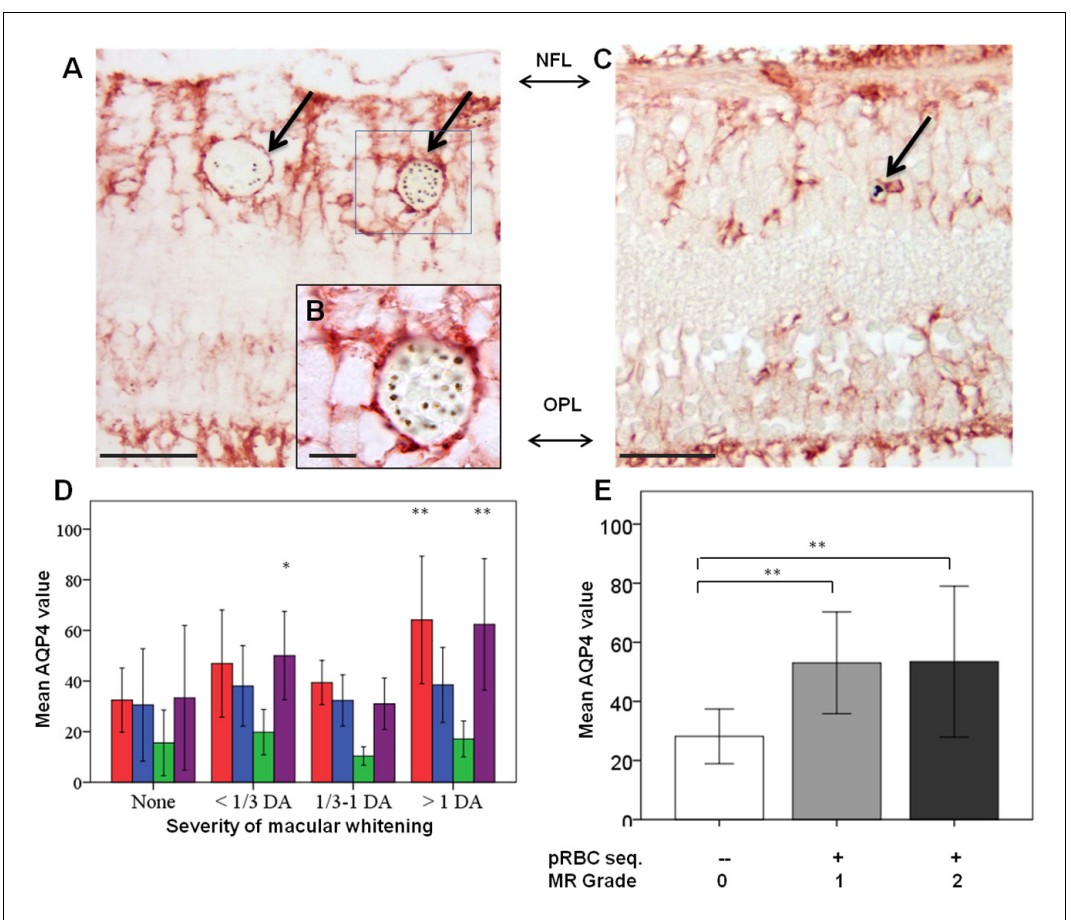

**Figure 7.** Clinicopathological association between retinal whitening in the macula and increased AQP4 expression in malarial retinopathy. (A–C) Immunostaining pattern in the macula with (A-B, case no 13) and without whitening (C, case no 21). Parasitised vessels are marked by arrows. The vertical linear pattern indicates Műller cell immunoreactivity for AQP4. (D) Cluster column chart showing densitometrically assessed intensity of immunoreactivity ('value') of AQP4 levels measured by IHC in the macula by retinal layers: nerve fibre layer = NFL (red), ganglion cell layer = GCL (blue), inner plexiform layer = IPL (green), outer plexiform layer = OPL (purple). (E): AQP4 levels in the nerve fibre layer plotted against MR severity classification groups (grade 0 = none, 1 = mild, two moderate/severe). Means ± SD are reported in all graphs; ANOVA was used to compare means (N = 26). *p<0.05 and **p<0.001. Scale bars: 50 µm (panels **C, E, F** and **G**); 10 µm (panel D).
DOI: https://doi.org/10.7554/eLife.32208.016

The following source data and figure supplements are available for figure 7:

**Source data 1.** Clinicopathological association between retinal whitening in the macula and increased AQP4 expression inmalarial retinopathy.
DOI: https://doi.org/10.7554/eLife.32208.018

**Figure supplement 1.** Clinicopathological association between retinal whitening in the peripheral retina and increased AQP4 expression in malarial retinopathy.
DOI: https://doi.org/10.7554/eLife.32208.017

**Figure supplement 1—source data 1.** Clinicopathological association between retinal whitening in the peripheral retina and increased AQP4 expression inmalarial retinopathy.
DOI: https://doi.org/10.7554/eLife.32208.019

## Quantitative image analysis of retinal sequestration

The results of our semi-quantitative image analysis to investigate the value of retinal sequestration to predict disease outcome are shown in *Figure 10*, including an example of the output from the algorithm (*Figure 10A*). Data were available on 251 eyes (one eye per case), and there were 33 (13.1%) deaths. The mean ratio of affected:unaffected vessels was 41.9% in children who died and 37.8% in survivors. The distribution of ratios across the 251 eyes is shown in *Figure 10B*; the amount of IVFD in retinal vessels was higher in the patients who died in our study, but the difference did not reach statistical significance (OR 18.05, 0.74–211.33, p<0.08).

## Discussion

The clinicopathological findings from our unique cohort provide strong evidence that the orange appearance of retinal vessels in comatose children with a clinical diagnosis of CM is caused by sequestered late-stage pRBCs. Our dataset of clinical outcomes, the largest to date, and our independently graded angiographic data show that this visible sequestration is strongly associated with death, with an increased risk when arterioles are involved. The tissue effects of sequestration are widespread within the neurovascular unit, including novel findings of severe loss/disruption of pericytes. Retinal whitening, also strongly associated with death, is associated with features of cytotoxic oedema, consistent with sequestration causing ischaemia.

We used three datasets to investigate if the features seen clinically in the retina represent sequestration, which is the principal underlying pathophysiological event in *P. falciparum* malaria. Our data from 817 children point definitively to the importance of sequestration seen clinically as visible orange vessels, associated with a 2.71-fold increased odds of death. Our data add to previous work by us (*Beare et al., 2004*) but with greater confidence and with specific reference to orange vessels rather than all retinal vessel abnormalities.

The orange colour of the sequestered intravascular material appears to be a result of a mix of sequestered late-stage pRBCs (containing haemozoin) adherent to the endothelium, surrounding a central narrowed blood column consisting of uninfected RBCs. Our numbers of cases and controls

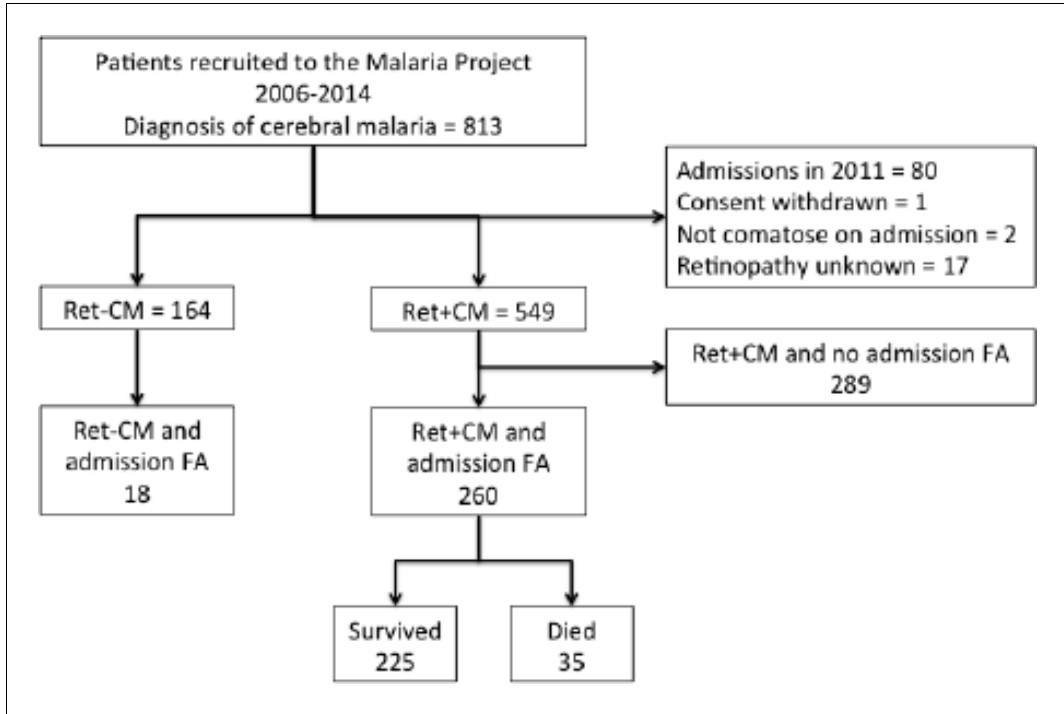

**Figure 8.** Flow chart describing fluorescein angiography dataset.
DOI: https://doi.org/10.7554/eLife.32208.020

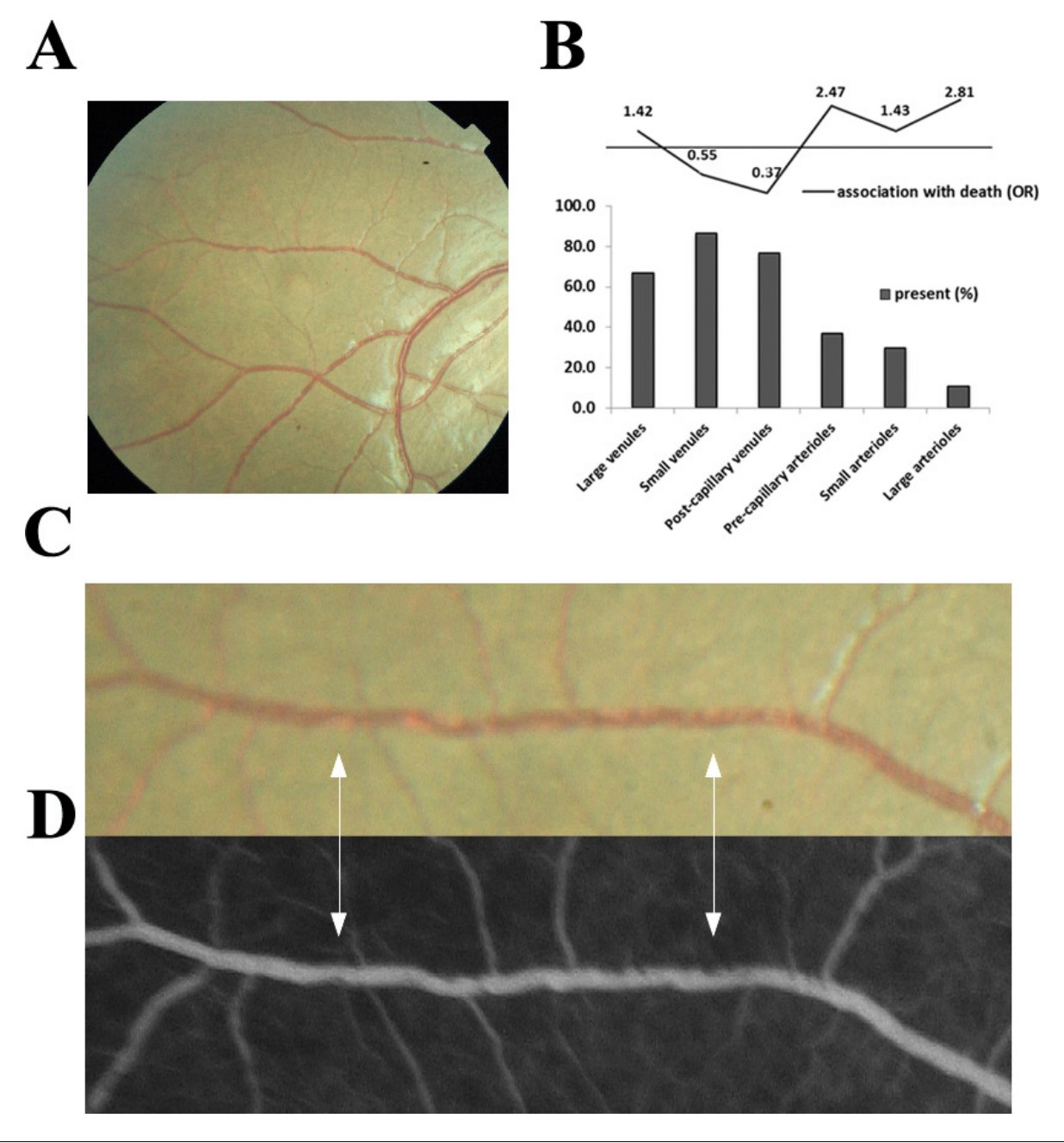

**Figure 9.** Visible sequestration in the retinal neurovasculature. (A–D): Orange intravascular material is seen in the retinal venule (**A, C**) which co-localises to the intravascular filling defects on fluorescein angiography (**D**) (see arrows). Chart (**B**) shows the frequency of visible sequestration in six microvessel types in 259 subjects with retinopathy +ve CM and the odds ratios of death within the admission.

DOI: https://doi.org/10.7554/eLife.32208.021

The following source data is available for figure 9:

**Source data 1.** Visible sequestration in the retinal neurovasculature.

DOI: https://doi.org/10.7554/eLife.32208.022

are typical for this type of pathological study and the numbers of vessels sampled were high. Our findings add to those reported by some of us previously (*Lewallen et al., 2000*), which described dehaemoglobinised RBCs in sequestration, by adding new topographical clinicopathology data. We found that sequestration involving 360° of the circumference of the vessel lumen was strongly

associated with the presence of orange discolouration clinically. We think orange vessels can be considered as an indication of severe sequestration and, as such, are clinically extremely valuable. This severe sequestration is easily visible with indirect ophthalmoscopy after pupil dilation. Less severe sequestration may be detectable with the newly available technology of hand-held optical coherence tomography (OCT) of the retina.

Retinal capillary involvement, in contrast to orange vessels, appears to be a phenomenon in CM not associated with death. Clinically, this is visible as white vessels and histologically predominantly as ruptured RBCs and extra-erythrocytic haemozoin, with no intact or non-parasitised RBCs. This feature was associated with non-perfusion on FA.

Our large fluorescein angiography (FA) study, which extended over eight seasons, shows that the retinal intravascular material was seen in nearly all MR-positive cases, especially the post-capillary and small venules (98.3% and 87.9% of gradeable vessels, respectively). These findings are novel, whereas our and others' earlier data have been descriptive. The limitations of imaging in comatose young children mean that our grading method was unable to reliably identify capillaries, and so we were unable to robustly investigate the capillaries angiographically. We believe that capillary involvement is typical of pRBC sequestration in the neurovasculature. The presence of intravascular material in the arterioles was much less likely (pre-capillary 58.4%, small 43.9%, large 15.3%). However, arteriolar intravascular material was associated with longer recovery times (p<0.01- < 0.02) and greater risk of death, with involvement of the large arterioles conferring a 2.81-fold increased risk of death. It appears that the involvement of the arteriolar side can be taken as a clinical marker of severity, indicating a greater extent or load of sequestration. We have previously described the features of intraretinal material coining the term 'intravascular filling defects'. This FA term can now be replaced by 'retinal sequestration'.

We identified an association between sequestration and profound changes in the cells of the retinal neurovascular unit. These cells are critical to the preservation of BRB function (*Kaur et al., 2008b*) and the changes have important parallels in the brain, especially for swelling (*Seydel et al., 2015*). Reduced expression of CD34 in endothelial cells and of SMA and PDGRFβ in pericytes indicates significant dysfunction of both cell types. Our pericyte data are novel; pericytes have not been extensively studied in malaria before, with only one study reporting pericyte vacuolation in adult fatal CM (*Pongponratn et al., 2003*). Reduction of SMA immunoreactivity may be related to two pathological mechanisms: vessel dilatation with altered pericyte function, or pericyte loss. PDGF-signalling is critical for the survival of endothelium in physiological conditions (*Armulik et al., 2005*). Pericytes are highly sensitive to hypoxia (*Kamouchi et al., 2012*), especially in brain and retina where they are most abundant, and when vessels lose or develop abnormal pericytes they become hyperdilated, show signs of vessel dysfunction, and haemorrhage may occur (*Bergers and Song, 2005*). Within the MR cases, we found more retinal vessels with abnormal pericyte staining in those cases presenting with retinal haemorrhages than those cases without. The retina and brain present similar pathological features in CM, including haemorrhages (*Greiner et al., 2015*). We found the same significant loss of pericytic SMA and PDGFRβ in a further small analysis comparing brain microvessels in the presence of pRBC sequestration (median %, min-max% of vessels with SMA intact: 15%, 9–20%; PDGFRβ: 13%, 6–24%) with non-parasitaemic vessels (SMA intact: 92%, 79–100%; PDGFRβ: 96%, 91–100%) (p<0.001 for all) (n = 5; Barrera V et al, unpublished). These data suggest that retina and brain may have similar dysfunction/loss of pericytes in fatal paediatric CM.

We also identified effects on astrocytes and Müller cells indicating wider effects on neural retinal cells than previously identified. Late reactive (*Hiscott et al., 1984*) GFAP was upregulated, but the greater effect was seen for the early-responsive (*Ortinski et al., 2010*) perivascular ICAM-1 perhaps reflecting the short survival time of children with fatal CM. Our group has also previously identified upregulation of β-amyloid precursor protein as evidence of axonal damage (*White et al., 2009*). These neuroglial effects of retinal sequestration are likely to be widespread and include disturbance of tight junction regulation causing BRB/BBB breakdown with vasogenic oedema, an implicated pathway for brain swelling and death (*Taylor et al., 2004*; *Mohanty et al., 2017*).

Retinal whitening is a key feature of MR. Our finding of whitening at the fovea conferring a 3.4-fold increased risk of death strengthens our previous findings (*Beare et al., 2004*). We have previously shown that retinal whitening is topographically associated with capillary non-perfusion and is found in watershed zones of the retina, sites of high metabolic demand (*Beare et al., 2009*), suggesting that tissue hypoxia is a principal pathogenic pathway (*White et al., 2009*).

Our immunohistochemistry data provide further evidence that the inner retina is affected by tissue hypoxia and intracellular oedema. Ganglion cells showed increased expression of VEGFR1, which, in combination with VEGF, is neuroprotective during ischaemia (*Saint-Geniez et al., 2008*). Glia in retinal zones where whitening is mainly localised were found to express AQP4, a water channel protein linked to hypoxic oncotic swelling. This observation is supported by our previous electrophysiologcal study, which showed abnormal B wave implicit time indicating inner retinal dysfuction in retinal whitening (*Lochhead et al., 2003*). This all supports inner retinal neuronal ischaemia as opposed to dysfunction of the outer retinal photoreceptors and choroidal circulation. Further studies with the OCT may shed new light on the retinal whitening.

We have some conflicting evidence on the importance of capillary non-perfusion (CNP). There are undoubted tissue effects of sequestration-induced hypoxia in the vessel and extending into the neuroretina causing tissue swelling and opacification. However, the whitening seen in capillaries was not associated with death, and sequestration seen in the post-capillary venules on FA, a frequent association with CNP, showed a trend for survival. Sequestration in post-capillary venules is more common than arterioles, and this suggests these children as a group were not as critically ill as those with sequestration extending additionally into arterioles.

So how can our findings affect the clinic management and future research directions in CM? The detection of orange vessels on clinical examination has a high sensitivity and specificity for a severe degree of sequestration, which is associated with death. Sequestration detectable on FA in the arterioles, and especially the large arterioles, is also predictive of death and probably indicates a high parasite load. Orange vessels can be seen clinically with the indirect and direct ophthalmoscope through a dilated pupil by a trained physician (*Taylor et al., 2004*), but these skills are mainly available in research or tertiary centres in malaria endemic areas (*Swamy et al., 2018*). We have with others recently developed MR detection algorithms offering a potential automated diagnostic tool for severe malaria in district hospitals (*Joshi et al., 2017*). Our new clinical markers of severe disease and poor outcome (visible orange vessels and arteriolar involvement indicating severe sequestration, and severe foveal whitening) should be a focus for diagnosis and management. It should be recognised that including children without MR in clinical trials is likely to reduce their power to detect an effect of an intervention on CM outcomes.

There is good evidence that the clinicopathological features of CM in the retina parallel those seen in the brain (*Barrera et al., 2015*; *MacCormick et al., 2014*): ring-shaped haemorrhages (*White et al., 2009*; *Dorovini-Zis et al., 2011*), pathology of sequestration, associations between retinal features

**Table 5.** Frequency of intravascular filling defects (worse eye) on fluorescein angiography manual grading by involvement of retinal vessel in 259 children with MR-positive disease and FA within 24 hr of admission and unadjusted association with death (n = 35) and coma recovery of consciousness (BCS ≥3; n = 225)

| Retinal vessel | Sequestration | Died* N | % | Total | Survived* N | % | Total | Association with death OR | 95% CI | p |
|---|---|---|---|---|---|---|---|---|---|---|
| large venules | present | 26 | 86.7 | 30 | 172 | 79.3 | 217 | 1.70 | 0.56–5.12 | 0.35 |
| | absent | 4 | 13.3 | | 45 | 20.7 | | | | |
| small venules | present | 29 | 96.7 | 30 | 211 | 98.1 | 215 | 0.88 | 0.71–1.09 | 0.23 |
| | absent | 1 | 3.33 | | 4 | 1.86 | | | | |
| post-capillary venules | present | 25 | 96.2 | 26 | 201 | 98.5 | 204 | 0.37 | 0.04–3.70 | 0.4 |
| | absent | 1 | 3.85 | | 3 | 1.47 | | | | |
| pre-capillary arterioles | present | 19 | 76.0 | 25 | 109 | 56.2 | 194 | 2.47 | 0.94–6.45 | 0.065 |
| | absent | 6 | 24.0 | | 85 | 43.8 | | | | |
| small arterioles | present | 15 | 51.7 | 29 | 93 | 42.9 | 217 | 1.43 | 0.66–3.11 | 0.37 |
| | absent | 14 | 48.3 | | 124 | 57.1 | | | | |
| large arterioles | present | 9 | 30.0 | 30 | 29 | 13.2 | 219 | 2.81 | 1.17–6.72 | **0.02** |
| | absent | 21 | 70.0 | | 190 | 86.8 | | | | |

DOI: https://doi.org/10.7554/eLife.32208.023

and that key neurological pathways seem to be non-functioning ('pathways to neural cell death'). Mendis K and others (*Mendis et al., 2009*) have argued that the long-term goal of eliminating malaria remains dependent on continuing research and the development of new drugs and therapeutic strategies to sustain control programmes. Better identification and treatment of severe malaria will also be needed. Our findings from manual and semiautomated image analysis provide an indication that quantification of the load of retinal sequestration is promising as a useful metric in clinical trials and merits further development to identify a severity cut-off.

The results we have presented in this paper from our long-term programme of research strongly support the concept that sequestration can be identified clinically in the retina at the bedside, and offer important new insights into the widespread effects of sequestration on the neural microvasculature and cells of the neurovascular unit. This sequestration can be seen in clinical practice at a critical time in the management of the comatose child in malaria endemic areas offering opportunities to study the effects of new therapies, as well as an early concrete diagnosis and a marker of severe disease.

# Materials and methods

**Key resources table** Antibodies used for immunohistochemistry analysis of the clinicopathology dataset

| Antigen | Specificity | MR feature | Manufacturer (clone); RRID[*] | Host[†] (class) | Ag retrieval[‡] | Dilution[§] | Chromogen[#] | Staining quantification | Ref |
|---|---|---|---|---|---|---|---|---|---|
| VEGFR1 | Retinal cell | Retinal whitening Tissue effects | Abcam (Y103); AB_778798 | Rb mAb (IgG) | Heat (High pH) | 1:2,000, 30 min RT | DAB | Automated | (*Kaur et al., 2008a*) |
| Aquaporin 4 (AQP4) | Neuroglia | Retinal whitening Tissue effects Intracellular oedema | Abcam (EPR7040); AB_11143780 | Rb mAb (IgG) | Heat (Low pH) | 1:500, 60 min RT | AEC | Automated | (*Medana et al., 2011*) |
| Glial fibrillary acidic protein (GFAP) | Neuroglia (late activation) | Vessel discolouration | Dako; AB_10013482 | Rb pAb | Proteinase K | 1:2,000, o.n. 4°C | AEC | Manual | (*Hiscott et al., 1984*) |
| ICAM-1 | Endothelium Neuroglia (early activation) | Vessel discolouration | Abcam (EP1442Y); AB_870702 | Rb mAb (IgG) | Heat (High pH) | 1:100, 30 min RT | DAB | Manual | (*Lee et al., 2000*) |
| CD61 | Platelets and precursors | Retinal whitening Vessel discolouration | Thermo Scientific; AB_929194 | Ms mAb (IgG1) | Heat (High pH) | 1:100, 32 min RT | DAB or AEC | Manual | (*White et al., 2009*) |
| CD34 (II) | Endothelium | Vessel discolouration | Dako (QBEnd-10); AB_2074478 | Ms mAb (IgG1k) | Heat (High pH) | 1:100, 30 min RT | DAB | Manual | (*Kaur et al., 2008a*) |
| Smooth muscle actin (SMA) | Pericyte (venules only) | Vessel discolouration | Dako (1A4); AB_2223500 | Ms mAb (IgG2ak) | Heat (Low pH) | 1:2,000, o.n. 4°C | AEC | Manual | (*Kaur et al., 2008b*) |
| Platelet derived growth factor receptor β (PDGFRβ) | Pericyte (signalling) | Vessel discolouration | Abcam (Y92); AB_777165 | Rb mAb (IgG) | Heat (Low pH) | 1:100, 30 min RT | DAB | Manual | (*Armulik et al., 2005*) |

[*]RRID: Research Resource Identifiers.

[†]Host: Rb = rabbit; Ms = mouse; mAb = monoclonal antibody; pAb = polyclonal antibody.

[‡]Ag retrieval: heat-mediated antigen retrieval was performed in high pH solution (10 mM Tris/1 mM EDTA, pH 9.0) or low pH solution (trisodium citrate 10 mM, pH 6.0). Proteinase K was from Dako (ready-to-use solution).

[§]Dilution and incubation time: RT = room temperature; o.n. = over night.

[#]Chromogen: AEC: 3-amino-9-ethylcarbazole; DAB = 3,3'-diaminobenzidine. Reported references are from main manuscript.

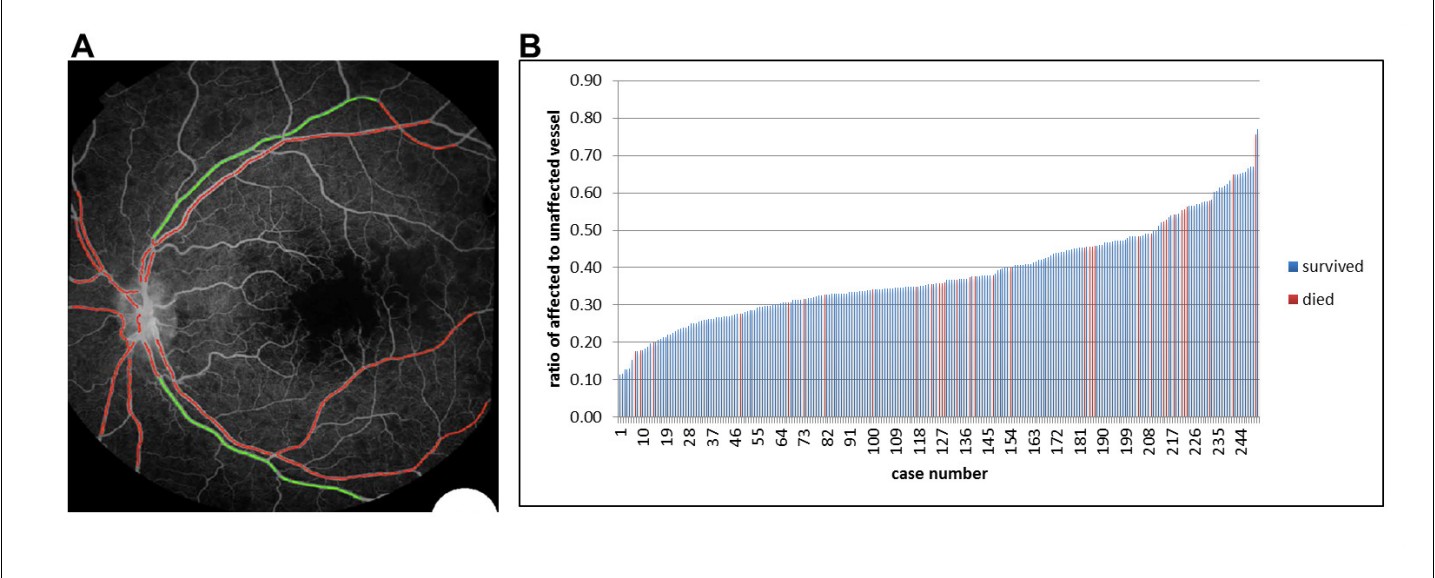

**Figure 10.** Semiautomated quantitative analysis of sequestration by length of affected vessel. (**A**) Example image of semiautomated system to show vessels affected by sequestration (red). (**B**) Chart showing distribution of proportion of detected vessel affected by sequestration related to survival in 251 eyes (one eye per case).

DOI: https://doi.org/10.7554/eLife.32208.024

The following source data is available for figure 10:

**Source data 1.** Semiautomated quantitative analysis of sequestration by length of vessel involved.

DOI: https://doi.org/10.7554/eLife.32208.025

## Study design and setting

A research programme based in Queen Elizabeth Central Hospital (QECH) and the College of Medicine in Blantyre, Malawi since 1996 provided the setting for the study. A prospective cohort of children (clinical dataset) was recruited between 1999 and 2014. A subcohort was selected for ocular histopathology (clinicopathology dataset, 1999–2011) and a second recruited for retinal photography (image analysis dataset, 2006–2014).

## Ethics

The core and specific studies all received approval from the research ethics committee at the University of Malawi College of Medicine P. 11/07/593, Michigan State University and the Royal Liverpool and Broadgreen University Hospital Trust n. 3690. Research was performed in accordance with the Declaration of Helsinki. Written consent for the clinical eye examination was sought in English or in the language of the parent/guardian who gave permission on the patient's behalf. If a patient died, additional informed written consent for autopsy was sought from the parent/guardian (*Taylor et al., 2004*; *Milner et al., 2013*).

## Subjects

### Clinical dataset

Children admitted to the Paediatric Research Ward of QECH with coma and suspected CM who met the definition of CM: presence of coma (Blantyre Coma Score (BCS) $\leq$2) and *P. falciparum* parasitaemia, in the absence of any other identifiable cause of coma (including meningitis, hypoglycaemia or postictal state of $\leq$2 hr) (*Taylor et al., 2004*). After initial stabilisation by the admitting paediatrics team, cases had pupils dilated and were examined by binocular indirect ophthalmoscopy with standardised data recording (*Harding et al., 2006*). Demographic, clinical and outcomes data (survival, death, time to recovery of consciousness (BCS $\geq$3)) were recorded and analysed (*Table 1*) after dual entry as previously described (*Seydel et al., 2015*). Peripheral parasitaemia, haemoglobin levels and HIV-1 serological status were determined as previously described (*Taylor et al., 2004*).

## Clinicopathology dataset

Clinicopathological cases were identified from an autopsy study performed between 1996 and 2010, which enrolled children who died of CM and parasitaemic children who died of other causes. Autopsy was performed to international standards within 12 hr of death. Clinical diagnosis of CM was from post-mortem brain analysis (*Taylor et al., 2004*). Specimens were obtained from the archive with: a full clinical eye examination performed during life, available severity grading of specific MR features, a clinical diagnosis of CM (see above) and evidence of valid consent (see below). Key pathology methods are given here, with further details available in Appendix 1.

Cases were allocated to three severity groups shown previously to reflect maturation stage and pigmentation of sequestered pRBC in the retinal capillaries and venules (*Barrera et al., 2015*):

- Grade 0 - pRBC sequestration 0–20% of retinal microvessels post mortem (and no extra-erythrocytic HZ deposition in retinal vessels), which also represents the cut-off value in the brain to confirm CM as the cause of death (*Taylor et al., 2004*)
- Grade 1 - pRBC sequestration in 20–60% of retinal microvessels and extra-erythrocytic HZ in ≤15% of retinal vessels
- Grade 2 - severe pRBC sequestration (>60% of retinal microvessels) and >15% contain extra-erythrocytic HZ (*Barrera et al., 2015*)

Eye specimens were anonymised, coded and, after fixation in 10% v/v neutral buffered formalin, processed as previously described (*Barrera et al., 2015*; *White et al., 2009*). Specimens were opened either horizontally in the pupil-optic nerve (PO) plane, or vertically. Retinal pathological features, such as orange/white vessel discoloration and intravascular material, were photographed and sampled using punch biopsies before wax embedding. Classification of the retinal zones used to compare levels of histological markers with severity of MR features detected during grading is described in Appendix 2.

All histopathological observations were performed masked to MR status. Up to 100 sequential sections were cut for each specimen and stained for H&E, Martius-Scarlet-Blue or immunohistochemistry. For detection of parasitic stage and elements in retinal vasculature, H&E stained sections were assessed for presence of pRBCs, and intra- and extra-erythrocytic HZ (*Barrera et al., 2015*). Percentages of capillaries and venules parasitised were calculated per MR grade (means ±SD reported): $6 \pm 5\%$ (grade 0); $54 \pm 12\%$ (grade 1); $87 \pm 16\%$ (grade 2).

Vascular endothelial growth factor receptor 1 (VEGFR1) and aquaporin-4 (AQP4) immunostaining were quantified by retinal layer, using a densitometry-based automated analysis method on eight randomly selected fields per section (see Appendix 1). For the vascular-related antigen markers (see Key Resources Table), the numbers of immunoreactive retinal vessels or segments were counted manually by one of the authors (VB) and at least one second independent observer (TF, SM or DG, see Acknowledgments). At least 100 capillaries and venules were analysed in each case and an inter-observer error count of less than 10% considered acceptable, otherwise a third observer assessed the case.

## Image analysis dataset

Children deemed by the admitting paediatrician to be sufficiently stabilised clinically underwent colour photography and FA following previously published protocols (*Beare et al., 2004*; *Harding et al., 2006*; *Beare et al., 2009*). Subjects were excluded if their guardians withdrew consent, if their clinical condition was deteriorating or rapidly improving to normal consciousness, or if the ophthalmologist was not available. A trained ophthalmologist graded the FA images against previously published protocols developed by the Liverpool Ophthalmic Reading Centre (*MacCormick et al., 2015*). Classification of retinal zones is described in Appendix 2 and used standardised validation procedures. The following were included: presence/absence, extent and distribution of whitening, vessel discolouration (divided into orange and white vessels as per analysis), haemorrhages and papilloedema. An automated segmentation algorithm was developed (method described elsewhere [*Zhao et al., 2015*]) to identify vessels with IVFD, applied to the macular image with best field definition and clarity from one eye of each case and analysed by proportion of vessel affected by IVFD/proportion not affected.

## Statistics

Relationships between clinical dichotomous outcome and studied variables was first analysed using simple logistic regression. To test for confounding, a multivariable logistic regression model was fitted within the clinical dataset, adjusting for variables significant at p<0.01 (*Table 1*) and including age. We did not include variables not fulfilling those described by Greenland et al. (*Greenland et al., 1999*): coma score (part of the causal pathway to death) and retinal haemorrhages (orange vessels can evolve to haemorrhages because of sequestration affecting vessel stability). Potential bias because of missing data was investigated by comparison between subjects examined and not examined. Coma recovery time was truncated at zero and highly skewed with over dispersion, and so we used truncated negative binomial regression to estimate unadjusted associations with this outcome. Clinicopathological correlation analyses used data from the last clinical examination before death and one eye per subject. After quantitative evaluations were completed, specimen codes were broken and results compared with the clinical data. Continuous scale data were assessed for normal distribution with the Shapiro-Wilk test. When normality was satisfied, one-way ANOVA (with Bonferroni post-hoc correction) was used to compare continuous scale data across MR severity groups, or retinal layers. Spearman correlation (with significance at p<0.01) was used to correlate a continuous scale variable with severity grades for macular and peripheral whitening. Fisher exact test was used to compare categorical variables (e.g. ICAM-1 or GFAP perivascular staining, discoloration presence/absence) and p values < 0.05 were considered significant after adjustment where appropriate for multiple comparisons. SPSS Statistics 22 was used throughout.

## Data availability

The anonymised datasets for this study – clinicopathology dataset (author: Valentina Barrera), clinical dataset (author: Ian MacCormick) and FA dataset (authors: Ian MacCormick and Yalin Zheng) – are stored at the University of Liverpool Research Data Management Archive (datasets archive created on 20/02/2018). Given the confidential nature of these data (clinical and histology images and clinical examination forms of the patients (with DOB, date of death, clinical parameters, cause of death), access is subject to reasonable request through the senior author, Simon P. Harding (sharding@liverpool.ac.uk), and to approval by the Malawi Malaria Consortium Data Oversight Committee (Terrie E. Taylor Director, Blantyre Malaria Project (ttmalawi@msu.edu) and SJ Gordon, Director and Chair Research Strategy Group, MLW Clinical Research Programme (sgordon@mlw.mw)).

## Acknowledgements

We thank the parents and guardians of the patients participating in the study. We thank Dr Simon J Glover for retinal examinations and data collection, and Drs Macpherson Mallewa, Dr Karl Seydel and the nurses of the Paediatric Research Ward at the Queen Elizabeth Central Hospital, Blantyre, Malawi, for caring for the patients. We thank Susan Lewallen for her contribution to the evolution of concepts within the manuscript. We thank Mr Tobi Fishpool, Miss Sohmal Musini and Mrs Duaha Ghafouri from University of Liverpool for their assistance with the histopathology quantitative analysis.

## Additional information

### Funding

| Funder | Grant reference number | Author |
| --- | --- | --- |
| Wellcome | #092668/Z/10/Z | Simon Peter Harding |
| NIH Clinical Center | #5R01AI034969-11 | Terrie Ellen Taylor |
| Wellcome | #074125 | Malcolm Edward Molyneux |

The funders had no role in study design, data collection and interpretation, or the decision to submit the work for publication.

## Author contributions

Valentina Barrera, Ian James Callum MacCormick, Conceptualization, Data curation, Software, Formal analysis, Investigation, Methodology, Writing—original draft, Project administration, Writing—review and editing; Gabriela Czanner, Conceptualization, Data curation, Software, Formal analysis, Validation, Methodology, Writing—review and editing, oversaw data quality and statistical analyses; Paul Stephenson Hiscott, Conceptualization, Data curation, Methodology, Writing—review and editing; Valerie Ann White, Conceptualization, Resources, Data curation, Formal analysis, Investigation, Methodology, Writing—review and editing; Alister Gordon Craig, Steve Kamiza, Conceptualization, Resources, Methodology, Writing—review and editing; Nicholas Alexander Venton Beare, Conceptualization, Data curation, Validation, Investigation, Methodology, Writing—review and editing; Lucy Hazel Culshaw, Formal analysis, Writing—review and editing; Yalin Zheng, Conceptualization, Data curation, Formal analysis, Methodology, Writing—review and editing; Simon Charles Biddolph, Data curation, Investigation, Writing—review and editing; Danny Arnold Milner, Conceptualization, Resources, Validation, Investigation, Methodology, Writing—review and editing; Malcolm Edward Molyneux, Conceptualization, Resources, Data curation, Funding acquisition, Methodology, Project administration, Writing—review and editing; Terrie Ellen Taylor, Conceptualization, Resources, Data curation, Supervision, Funding acquisition, Validation, Investigation, Methodology, Writing—review and editing; Simon Peter Harding, Conceptualization, Resources, Data curation, Formal analysis, Funding acquisition, Validation, Investigation, Writing—original draft, Project administration, Writing—review and editing

## Author ORCIDs

Valentina Barrera http://orcid.org/0000-0003-0515-5901
Simon Peter Harding http://orcid.org/0000-0003-4676-1158

## Ethics

Human subjects: The core and specific studies all received approval from the Research Ethics Committee at the University of Malawi College of Medicine P. 11/07/593, Michigan State University and the Royal Liverpool and Broadgreen University Hospital Trust n. 3690. Research was performed in accordance with the Declaration of Helsinki. Written consent for the clinical eye examination was sought in English or in the language of the parent/guardian who gave permission on the patient's behalf. If a patient died, additional informed written consent for autopsy was sought from the parent/guardian.

## Decision letter and Author response

Decision letter https://doi.org/10.7554/eLife.32208.035
Author response https://doi.org/10.7554/eLife.32208.036

## Additional files

### Supplementary files

• Supplementary file 1. Comparison of children without and with admission retinal exam data.
DOI: https://doi.org/10.7554/eLife.32208.027

• Transparent reporting form
DOI: https://doi.org/10.7554/eLife.32208.028

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

## Appendix 1

DOI: https://doi.org/10.7554/eLife.32208.029

## Supplementary pathology methods

Pupil-optic (PO) nerve wax blocks were cut into sections, and H&E stained to identify retinal areas for topographical correlation and subsequent histopathology (*Appendix 1 Figure 1*).

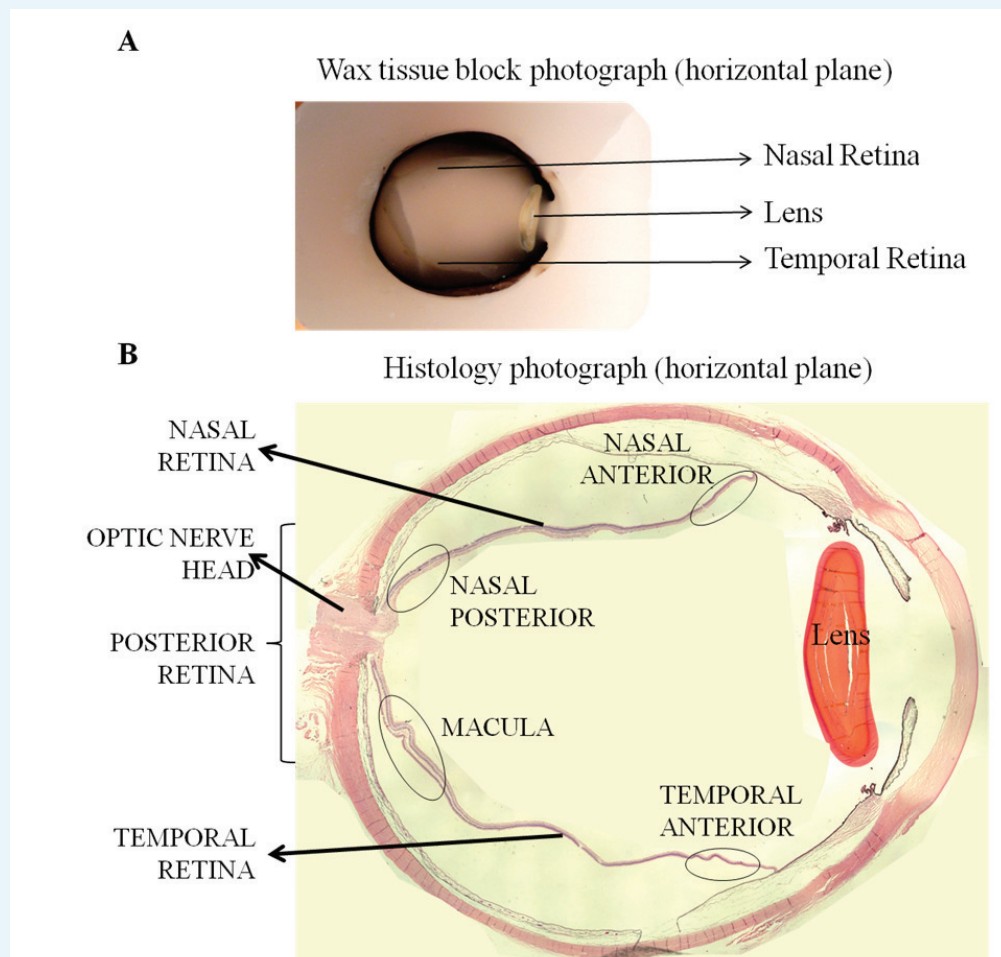

**Appendix 1—figure 1.** Orientation and topographical association in whole eye histology blocks (**A**) and in eye sections (**B**), used to perform correlation studies between fundal images and histology.

DOI: https://doi.org/10.7554/eLife.32208.030

Histology photographs were taken from randomly selected fields in each retinal area (macula, nasal posterior, nasal and temporal periphery) in sequential sections stained for immunohistochemistry markers, and used to measure marker intensity (see VEGFR1 and AQP4 analyses).

The macula is clearly identifiable histologically, due to a higher density of ganglion cell nuclei compared to other retinal areas (see section below). Optic nerve head and optic disc were used as matching references in histological and clinical photographs respectively. The retinal area on the nasal side of the optic nerve head (nasal posterior, also defined as near periphery; *Appendix 2—figure 1* panel B) corresponding to retinal zone 1 in the periphery (Appendix 2). Nasal and temporal anterior areas were considered matches for zone 2–3.

## Gross pathology

Eyes were examined macroscopically in 70% v/v ethanol with a dissecting microscope and orange/white discoloration of retinal vessels, intravascular material and retinal haemorrhages were recorded photographically. Punch biopsies (N = 4, see *Table 2*, main manuscript) were performed *post-mortem* to obtain individual retinal lesions. Calottes were also used to sample individual retinal features, after sectioning into small tissue strips (N = 7).

Tissue samples were dehydrated and embedded in paraffin wax. Sections, 3–4 µm thick, were cut with a manual rotary microtome for staining.

## Immunohistochemistry and microscopic pathology

Sections were deparaffinised, rehydrated and stained with standard hematoxylin-eosin (H&E) or with the indirect immunoperoxidase technique (see Key Resources Table for antigen retrieval treatment and list of antibodies). Endogenous peroxidases and non-specific binding were blocked by treating rehydrated sections with 0.3% v/v hydrogen peroxide (15 min; Dako) and 20% v/v goat serum (Sigma Aldrich) respectively. Ready-to-use Dako EnVision™ + System HRP was used for immunostaining (Key Resources Table). Anti-rabbit-HRP and anti-mouse-HRP secondary antibodies were incubated for 30 min. Negative and positive control experiments were run in parallel using, respectively, isotype control antibodies on retinal samples or tonsil. Other ocular tissues, such as optic nerve, choroid and ciliary body, were used as internal positive or negative controls. Microscopic investigations were carried out with an Olympus BX60 system microscope. Images were taken with an Olympus DP71 microscopic digital camera and cell imaging software (Olympus).

## Retinal layers

The retina is customarily divided into ten layers identifiable on H&E stained light microscopy (*Appendix 1—figure 2*).

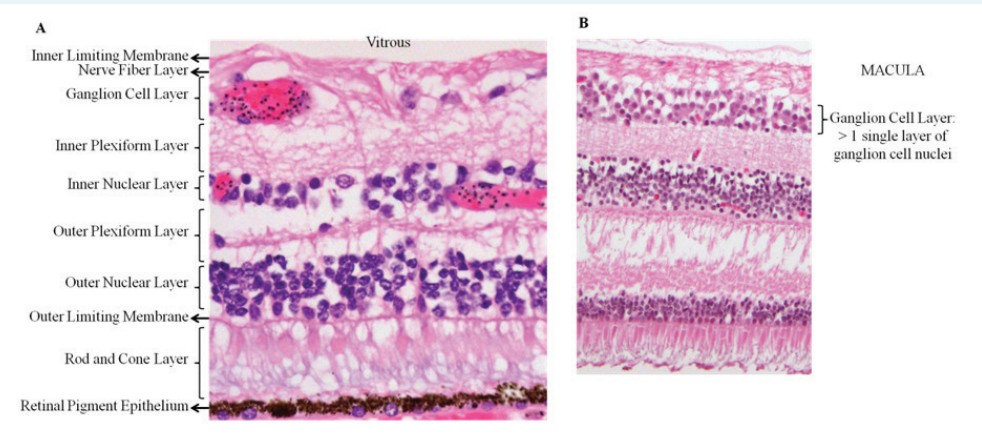

**Appendix 1—figure 2.** Panel A: retinal structure on light microscopy (H&E staining). Panel B shows the specific feature of >1 cell thickness in the ganglion cell layer, used to identify the macula.
DOI: https://doi.org/10.7554/eLife.32208.031

Retinal layers from inner to outer are: inner limiting membrane, nerve fibre layer (NFL), ganglion cell layer (GCL), inner plexiform layer (IPL), inner nuclear layer (INL), outer plexiform layer, outer nuclear layer (ONL), outer limiting membrane, photoreceptor outer segments (rod and cone), retinal pigment epithelium. The retinal neurovasculature is localised in the GCL and INL, with a capillary network in each and it forms the inner blood retinal barrier (BRB) comprising endothelial tight junctions and maintained by additional perivascular cells (astrocytes, Müller cells and pericytes. RPE tight junctions form the outer BRB. The macula is identified in histological sections by the presence of more than one ganglion cell nucleus in the GCL (*Appendix 1—figure 2B*).

## Immunohistochemistry (IHC)

The antibodies used to investigate the tissue effects of intravascular material are listed in Key Resources Table. IHC staining was quantified per retinal layer in each image by Image J 1.49 v (NIH, http://rsb.info.nih.gov/ij/). RGB images were converted to grey scale images without changing brightness or contrast, and regions of immunolabelling were selected by density thresholding. Low and high thresholds were selected by comparison of the staining intensity on similar sections from MR negative cases, and the thresholds were kept constant between cases for each marker using internal standards. Data were reported as area of microphotographs covered by the immunolabelling, normalised against the background (eye vitreous intensity).

## Appendix 2

DOI: https://doi.org/10.7554/eLife.32208.032

### Classification of retinal zones in grading of malarial retinopathy

Definitions of retinal zones for grading of clinical photographs and for the topographical clinicopathological study are shown in the Figure below.

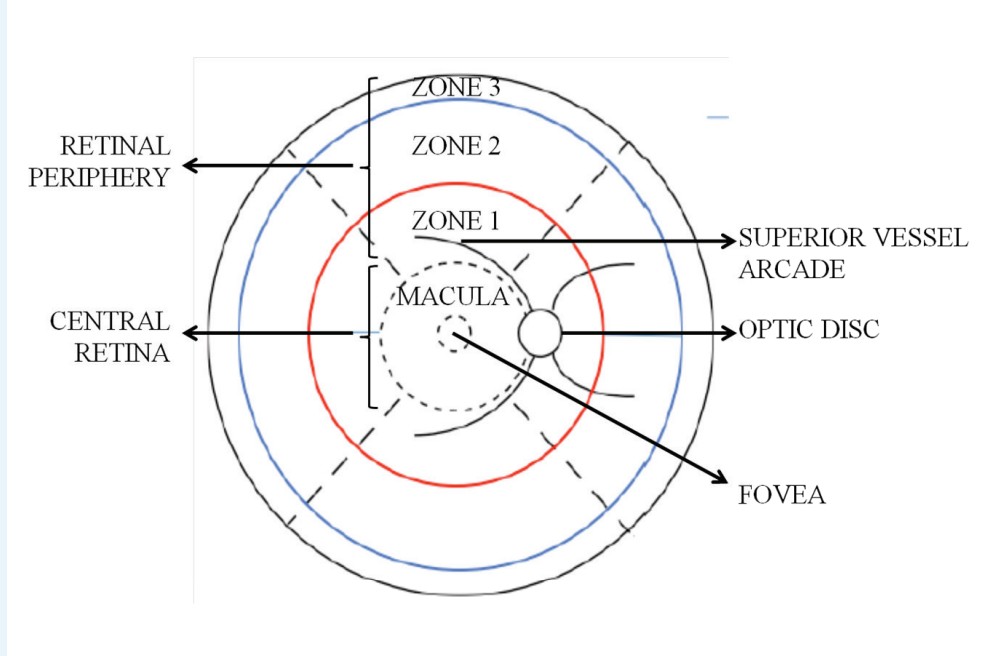

**Appendix 2—figure 1.** Retinal zones used for clinical grading.

DOI: https://doi.org/10.7554/eLife.32208.033

**Macula**: defined as the zone of retina within a circle centred on the centre of the fovea, which is the central retinal area with highest photoreceptor density. Macular boundaries are defined by vessels arcades.

**Peripheral retina**: defined as all retinal tissue lying outside the macular borders, divided into quadrants (temporal, superior, nasal, inferior) which are all graded separately during ophthalmoscopy. Gradeable peripheral retina was measured using zones (zones 1–3).

**Retinal whitening** was graded separately from the MR grade, yielding four severity grades. In order to assess the extent of macular involvement in whitening, macular zones of involvement were compacted into a notional circle using the optic disc as the nominal equivalent of a disc area (DA). Macular whitening severity grades are: none,<1/3 DA, 1/3–1 DA and >1 DA.

**Peripheral whitening** was also graded into four categories: none, Grade 1, Grade 2 and Grade 3 for each retinal quadrant, with a summation score to allow for the possibility of one or more quadrants being unobservable.

