## [Decision Letter]

Thank you for submitting your article "Neurovascular sequestration in paediatric *P. falciparum* malaria is visible clinically in the retina" for consideration by *eLife*. Your article has been favorably evaluated by Prabhat Jha (Senior Editor) and three reviewers, one of whom served as a guest Reviewing Editor. The following individual involved in review of your submission has agreed to reveal his identity: Charles Newton.

Summary:

Overall this is a well written, comprehensive clinical review by leaders in the field with new insights into the clinical pathology and pathogenesis of neurovascular sequestration in malarial retinopathy. Many of the conclusions are reconfirming previous clinical observations and outcomes the study team has already published on but on a larger sample size. The associated clinicopathological evaluations are firsts in describing the pathology of the retinal findings, though only a limited number of unique samples were assessed. The FA and immunochemistry results further delineate and expand on previous findings by the group but with new novel findings on a cellular and molecular level.

The reviewers have discussed the reviews with one another and the Reviewing Editor has drafted this decision to help you prepare a revised submission.

Essential revisions:

1) The clear delineation of what work was presented in this manuscript is an extension or summary of previous work published versus new novel research being presented for the first time in this publication.

2) Retinal findings and associated conclusions regarding these findings require additional consideration and analyses, including multivariant analyses, correlation with brain findings (imaging/tissue markers), impact on clinical diagnostic criteria, and further identification and assessment of additional relevant variables/confounders.

3) In general, the manuscript comes across a bit overwhelming in the breadth of information provided and is difficult to follow as a cohesive narrative. Further clarity on what populations were used for what analyses (and again if this is new or an expansion of previously published data) is needed as well as improvements on how all the findings presented diverge into a common narrative at the conclusion of the manuscript.

---

## [Author Response]

1) The clear delineation of what work was presented in this manuscript is an extension or summary of previous work published versus new novel research being presented for the first time in this publication.

We recognise that this publication reports research on a cohort of patients from whom data has been published previously by us and others in our group. The dataset is large, complex and unique and has been added to over 15 years to address complex questions as they have emerged. In the case of the work presented in this publication we set out to understand sequestration and its relationship to clinically identifiable features on ophthalmoscopy. The focus on ophthalmoscopy is important because it opens up opportunities to improve clinical management of comatose children with advanced CM.

Our previous work on vessel changes and whitening has described the clinical appearance but not established the aetiology. In Beare et al., 2004 we gave descriptive data and uni- and multivariate analyses of 278 clinical examinations performed between 1999 and 2003. We identified “vessel changes” and retinal whitening as associated with death (OR: 2.4, 2.0 respectively). In the publication under review now we extended our previous analysis of 5 years’ clinical data to 15 years to confirm the clinical importance of specific features of MR and identified orange vessels and foveal whitening as important predictors of death. We used these findings to design our clinicopathological study.

The histological data presented in this manuscript is presented for the first time. We, with others, have published four previous papers on the retinal histopathology of CM. Lewallen et al. (2000) (n=6) and White et al. (2009) (n=64) reported descriptive and correlation data from a different cohort from the same autopsy study, describing sequestration in retinal vessels and identifying a relationship with death from CM. Some of the cases in the White 2009 dataset were used in constructing a new dataset for the novel topographical clinicopathology study that we present in this manuscript. Our previous paper from this new cohort of cases (Barrera et al., 2015) reported differential sequestration rates in different ocular tissues. The paper from Griener et al studied brain-retina correlations in haemorrhage, ganglion cell axons and barrier integrity. All these studies reported sequestration in retinal vessels but without clinicopathological or topographical correlation. The clinical relevance of the features visible on clinical examination has not been well established.

In Beare et al., 2009 we gave descriptive data on the images from fluorescein angiography of 34 cases in the 2006 season. We described the intravascular material but its cause and significance remained unknown. Since then we have recruited a further 226 children to create our image analysis dataset allowing association studies that have not been feasible previously and data from semiautomated image analysis that is novel.

We have reviewed the descriptions of the datasets and have restructured the final paragraph in the Introduction and the first paragraph in the Materials and methods section. We have taken the opportunity to try to improve the clarity of the dataset descriptors and the study design.

Introduction: “We with other colleagues have previously reported descriptive pathological investigations of the features of MR (Lewallen et al., 2000) including clinical associations (White et al., 2009) and suggesting mechanisms. […] The further analysis of our clinical dataset is an extension of our previous association study while all other analyses presented in this manuscript are new”.

Materials and methods:

Study design and setting

“A research programme based in Queen Elizabeth Central Hospital (QECH) and the College of Medicine in Blantyre, Malawi since 1996 provided the setting for the study. A prospective cohort of children (clinical dataset) was recruited between 1999 and 2014. A subcohort was selected for ocular histopathology (clinicopathology dataset, 1999 – 2011)and a second recruited for retinal photography (image analysis dataset, 2006-2014)”.

2) Retinal findings and associated conclusions regarding these findings require additional consideration and analyses, including multivariant analyses, correlation with brain findings (imaging/tissue markers), impact on clinical diagnostic criteria, and further identification and assessment of additional relevant variables/confounders.

Our team has carefully considered these important comments as suggested and give our responses below. We have conducted multivariable analyses, and some additional experiments to add further insight into correlation with brain histopathology for our new findings on tissue effects of sequestration.

Multivariant analyses and confounders:

We performed multivariable analyses of the clinical dataset. We reviewed potential confounders which showed a significant univariate association with death at the level of p<0.01 (see Table 1 unadjusted associations) and where they do in our view fulfill the confounder criteria described by Greenland, Pearl and Robins, 1999. We did not include two variables that we believe do not meet Greenland’s criteria: coma score (part of the causal pathway to death) and retinal haemorrhages (orange vessels can evolve to haemorrhages due to sequestration affecting vessel stability). We included white cell count, lactate, HRP2, papilloedema as potential confounders. We also included age as it has been previously identified as a potential confounder in other paediatric CM studies.

We ran two separate regression analyses of outcome=death for the two ophthalmoscopic features of MR that are the focus of our study, as mentioned in the Introduction: orange vessels and foveal whitening. Both of the variables remained significant (p<0.01) and odds ratios were similar.

Both analyses add further evidence of the association of orange vessels and foveal whitening with death. Both analyses reported above were run on data from the worse eye, but significance did not change when two eyes were assessed separately (not shown). We have added brief statements on these results, and related methodology, in the revised manuscript.

Results: “The presence of visible orange vessels on ophthalmoscopy (Figure 1C-D) was significantly associated with death (OR 2.90, 1.96-4.30, p<0.001) as was severe foveal whitening (>2/3 foveal area; OR 3.40, 1.80-6.30, p<0.001 in simple logistic regression – see Table 1). When including potential confounders (age, WCC, HRP2, lactate, papilloedema (see Materials and methods)) in a multivariable regression model for the presence of the two retinal features for death, we found similar ORs and significance (orange vessels: OR 2.85, 1.72-4.74, p<0.001, n=549; foveal whitening: OR 3.57, 1.57-8.13, p=0.002, n=615).”

Materials and methods: “Relationships between clinical dichotomous outcome and studied variables was first analysed using simple logistic regression. […] We did not include variables not fulfilling the confounder criteria described by Greenland et al. (Greenland, Pearl and Robins, 1999): coma score (part of the causal pathway to death) and retinal haemorrhages (orange vessels can evolve to haemorrhages due to sequestration affecting vessel stability)”.

Correlation with brain findings:

We acknowledge the importance of comparing retinal and brain findings. This paper is focused on the retina and we believe that adding in analyses on brain imaging and tissue markers will greatly lengthen the paper. However we have used this opportunity to review the discussion on the retina/brain correlates undertaken some additional brain histology to extend our new findings in the retina on pericytes.

We have added in data on pericyte markers (SMA and PDGFRβ) in cerebral cortex sections from 5 cases with or without MR positive CM. The cases were from the patient cohort included in our manuscript, thus data corresponding to retinal staining are available. A minimum of 50 parasitised and non-parasitised microvessels were counted per case, by 2 independent observers (VB and LHC). In these brain samples, we found a significant reduction of normal pericytic SMA and PDGFRβ staining, in the presence of pRBC sequestration (median% , min-max% of vessels with SMA intact: 15%, 9-20%; PDGFRβ: 13%, 6-24%), compared to non-parasitemic vessels (SMA intact: 92%, 79-100%; PDGFRβ: 96%, 91-100%) (p<0.001 for all). These proportions are very similar to those observed in MR positive and MR negative cases. This suggests that retina and brain present a similar dysfunction/loss of pericytes in fatal paediatric CM.

Discussion: “Pericytes are highly sensitive to hypoxia (Kamouchi et al., 2012), especially in brain and retina where they are most abundant, and when vessels lose or develop abnormal pericytes they become hyperdilated, show signs of vessel dysfunction and may haemorrhage (Bergers and Song, 2005). […] These data suggest that retina and brain present a similar dysfunction/loss of pericytes in fatal paediatric CM”.

Impact on clinical diagnostic criteria:

We considered again the impact of our findings on the clinical criteria used in the diagnosis of CM. The focus of our publication is not on the primary diagnosis of CM but on the previous uncertainty of the role of sequestration in the features of malarial retinopathy, as well as the identification of those comatose children who can progress to advanced and fatal CM. The identification that orange material is sequestration adds strength to the importance of the previously poorly understood “vessel changes” in making the diagnosis. The separation from white vessels may also prove helpful. However our study presents some limitations as the studied cohort does not include comatose children without CM. We have added some emphasis to paragraph eleven of the Discussion.

Discussion: “Our new clinical markers of severe disease and poor outcome (visible orange vessels and arteriolar involvement indicating severe sequestration, and severe foveal whitening) should be a focus for diagnosis and management”.

3) In general, the manuscript comes across a bit overwhelming in the breadth of information provided and is difficult to follow as a cohesive narrative. Further clarity on what populations were used for what analyses (and again if this is new or an expansion of previously published data) is needed as well as improvements on how all the findings presented diverge into a common narrative at the conclusion of the manuscript.

We recognise that the content of our manuscript is extensive. As stated in our covering letter we decided to study 3 datasets from our longstanding research programme designed to provide sufficiently new and definitive evidence in response to the comments from our submission of the histopathological data in 2015. We have tried to offset the large amount of data by concentrating only on sequestration and its tissue effects. We have addressed and clarified the populations studied in our response to point 1.